# Ranking-based Preference Optimization
# for Diffusion Models from Implicit User Feedback

**Yi-Lun Wu**     **Bo-Kai Ruan**     **Chiang Tseng**     **Hong-Han Shuai**
Institute of Electrical and Computer Engineering, National Yang Ming Chiao Tung University
{yilun.ee08,bkruan.ee11,chiang.ee11,hhshuai}@nycu.edu.tw

## Abstract

Direct preference optimization (DPO) methods have shown strong potential in aligning text-to-image diffusion models with human preferences by training on paired comparisons. These methods improve training stability by avoiding the REINFORCE algorithm but still struggle with challenges such as accurately estimating image probabilities due to the non-linear nature of the sigmoid function and the limited diversity of offline datasets. In this paper, we introduce Diffusion Denoising Ranking Optimization (Diffusion-DRO), a new preference learning framework grounded in inverse reinforcement learning. Diffusion-DRO removes the dependency on a reward model by casting preference learning as a ranking problem, thereby simplifying the training objective into a denoising formulation and overcoming the non-linear estimation issues found in prior methods. Moreover, Diffusion-DRO uniquely integrates offline expert demonstrations with online policy-generated negative samples, enabling it to effectively capture human preferences while addressing the limitations of offline data. Comprehensive experiments show that Diffusion-DRO delivers improved generation quality across a range of challenging and unseen prompts, outperforming state-of-the-art baselines in both both quantitative metrics and user studies. Our source code and pre-trained models are available at https://github.com/basiclab/DiffusionDRO.

## 1   Introduction

Text-to-image diffusion models have recently emerged as a powerful class of generative models, achieving impressive results in synthesizing high-fidelity images from textual descriptions [37, 27, 33, 5, 32, 22]. These models use iterative denoising to progressively transform random noise into coherent visuals aligned with the input text [13]. Despite their capabilities, users often expect outputs that not only match the text but also reflect implicit aesthetic or stylistic preferences that are hard to encode explicitly. As a result, aligning these models with nuanced human preferences has become an emerging challenge.

Existing approaches to preference alignment in generative models have predominantly relied on reinforcement learning frameworks, such as Reinforcement Learning from Human Feedback (RLHF) [6, 2, 9, 3, 16, 28]. In these methods, models are fine-tuned using reward signals derived from human evaluations, often requiring paired datasets where one output is deemed better than another. Methods such as Direct Preference Optimization (DPO) have been employed in large language models (LLMs) and diffusion models to optimize for human preferences effectively [30, 35, 24].

However, the necessity for paired comparative data introduces substantial practical limitations. Collecting this data is labor-intensive and time-consuming, and it might not encompass the full spectrum of user preferences, potentially omitting users' favored choices. Moreover, even when successfully obtained, paired data may not effectively optimize for user preferences. For example, differentiating between two high-quality options may not yield meaningful insights for preference

39th Conference on Neural Information Processing Systems (NeurIPS 2025).

determination, as both already meet the desired criteria. Conversely, comparing two poor examples fails to provide the model with positive references needed to avoid undesirable features. In both cases, the comparative data may be too limited to enable the model to discern and learn the nuances that truly align with human preferences, potentially resulting in inconsistent or suboptimal outputs.

In this paper, we propose a novel approach for fine-tuning diffusion models to optimize user preferences using only demonstration examples—images that embody the desired qualities—without comparing them to less preferred outputs. This method departs from traditional approaches reliant on paired data, where examples are ranked against each other. Instead, we use these positive examples as direct guides, teaching the model to produce outputs that reflect the desired attributes. By theoretically deriving a novel optimization objective, we enable the diffusion model to learn from these demonstration examples while mitigating the risk of overfitting.

The contributions can be summarized as follows.

- We introduce a novel framework for preference optimization in diffusion models that requires expert demonstrations only, addressing the limitations of methods that depend on paired comparative data.

- We derive an optimization objective that compares the model's outputs with the demonstration examples. This formulation ensures effective learning from positive examples while preventing overfitting.

- Through extensive experiments, we demonstrate that our method achieves a preference rate exceeding 70% in terms of PickScore compared to state-of-the-art models. Our approach not only better aligns with desired human preferences but also exhibits robustness and strong generalization to unseen data.

## 2 Related Work

To guide diffusion models toward preferred outcomes, Reinforcement Learning (RL) has been widely adopted, including DDPO [3] and TDPO-R [42], which apply REINFORCE to optimize generation trajectories based on human feedback. However, these methods often suffer from complex reward design and high variance, leading to unstable training [8]. To address these challenges, Fan et al. [9] propose DPOK, introducing a KL divergence term to penalize deviation from the base model and prevent reward hacking [10]. Similarly, PRDP [8] uses a distillation-like strategy where a reward model predicts preferences to guide diffusion model updates, though it remains reliant on the reward model's accuracy.

Recent work has also explored Direct Preference Optimization (DPO) in diffusion settings. D3PO [39] and Diffusion-DPO [35] adapt DPO techniques from LLMs to fine-tune diffusion models directly from preference-paired data without a separate reward model. Diffusion-KTO [17] further simplifies supervision by decoupling preference pairs into binary positive/negative sets, though this can introduce semantic biases—e.g., if negative sets are skewed toward specific concepts such as cats, the model may learn undesirable associations.

While RL and DPO-based methods advance preference alignment, they heavily rely on large-scale paired data. SPIN-Diffusion [41] circumvents this by leveraging earlier model checkpoints as negative samples, enabling self-improvement from positive-only data. However, its multi-stage pipeline demands careful hyperparameter tuning to ensure consistent performance.

In contrast, we formulate preference alignment as a max-margin inverse reinforcement learning (IRL) problem, deriving a single-stage objective that encourages the generation of high-quality samples. This formulation eliminates the need for negative samples, thereby reducing semantic bias to some extent. Moreover, it provides a more stable and interpretable training process without requiring iterative self-play or stage-wise tuning.

## 3 Method

### 3.1 Background

We begin with a brief overview of the reinforcement learning framework and the foundational equations for our subsequent derivations. Specifically, for a given image-text pair $(\boldsymbol{x}_0, \boldsymbol{c})$, there exists an optimal reward model $r(\boldsymbol{x}_0, \boldsymbol{c})$ that assigns a score representing the level of human preference for the provided image-text pair. Building on prior work [9, 8, 35, 39], the objective of reinforcement learning from human preferences, as defined by the reward model, is formulated as follows:

$$\max_{p_{\boldsymbol{\theta}}} \mathbb{E}_{\boldsymbol{c} \sim \mathcal{C}, \boldsymbol{x}_0 \sim p_{\boldsymbol{\theta}}(\boldsymbol{x}_0 | \boldsymbol{c})} \Big[ r(\boldsymbol{x}_0, \boldsymbol{c}) \Big] - \beta \mathbb{D}_{\mathrm{KL}} \Big[ p_{\boldsymbol{\theta}}(\boldsymbol{x}_0 | \boldsymbol{c}) \| p_{\boldsymbol{\theta}_{\mathrm{ref}}}(\boldsymbol{x}_0 | \boldsymbol{c}) \Big], \tag{1}$$

where $\mathcal{C}$ represents the set of prompts, $\beta$ is the regularization weight, and $p_{\boldsymbol{\theta}_{\mathrm{ref}}}$ denotes the reference distribution, typically provided by a pre-trained diffusion model. As shown in prior work [11, 15, 25, 26], the optimal density function for Eq. (1) can be derived as:

$$p_{\boldsymbol{\theta}}^*(\boldsymbol{x}_0 | \boldsymbol{c}) = \frac{1}{Z(\boldsymbol{c})} p_{\boldsymbol{\theta}_{\mathrm{ref}}}(\boldsymbol{x}_0 | \boldsymbol{c}) \exp\left( \frac{1}{\beta} r(\boldsymbol{x}_0, \boldsymbol{c}) \right), \tag{2}$$

where $Z(\boldsymbol{c}) = \int_{\boldsymbol{x}_0} p_{\boldsymbol{\theta}_{\mathrm{ref}}}(\boldsymbol{x}_0 | \boldsymbol{c}) \exp\left( \frac{1}{\beta} r(\boldsymbol{x}_0, \boldsymbol{c}) \right) \mathrm{d}\boldsymbol{x}_0$ is the partition function. Through algebraic manipulation, the reward function can be reformulated as:

$$r(\boldsymbol{x}_0, \boldsymbol{c}) = \beta \log \frac{p_{\boldsymbol{\theta}}^*(\boldsymbol{x}_0 | \boldsymbol{c})}{p_{\boldsymbol{\theta}_{\mathrm{ref}}}(\boldsymbol{x}_0 | \boldsymbol{c})} + \beta \log Z(\boldsymbol{c}). \tag{3}$$

Nevertheless, calculating the probability of a clean image in a diffusion model requires marginalizing over the joint distribution $p_{\boldsymbol{\theta}}(\boldsymbol{x}_0 | \boldsymbol{c}) = \int_{\boldsymbol{x}_{1:T}} p_{\boldsymbol{\theta}}(\boldsymbol{x}_{0:T} | \boldsymbol{c}) \mathrm{d}\boldsymbol{x}_{1:T}$. This approach necessitates back-propagation through time [7, 38] to update the model, leading to a substantial increase in memory requirements for training. To mitigate this, we propose a reward model for the entire denoising trajectory, which enables a stepwise gradient calculation to improve computational efficiency.

### 3.2 Trajectory Reward Modeling

Conventional reward models typically predict preference scores solely for the final clean image $\boldsymbol{x}_0$, without considering the denoising trajectory. To decompose the full denoising process into individual steps, we follow prior work [35, 42] and assume the existence of a trajectory reward model $R(\boldsymbol{x}_{0:T}, \boldsymbol{c})$ for a diffusion model $p_{\boldsymbol{\theta}}(\boldsymbol{x}_{0:T} | \boldsymbol{c})$ such that:

$$r(\boldsymbol{x}_0, \boldsymbol{c}) = \mathbb{E}_{\boldsymbol{x}_{1:T} \sim p_{\theta}(\boldsymbol{x}_{1:T} | \boldsymbol{x}_0, \boldsymbol{c})} \big[ R(\boldsymbol{x}_{0:T}, \boldsymbol{c}) \big], \tag{4}$$

where $r(\boldsymbol{x}_0, \boldsymbol{c})$ aligns with human preferences as defined in Eq. (1). By substituting the reward function in Eq. (4) into Eq. (1) and applying the data processing inequality to expand the KL divergence, we have:

$$\mathbb{E}_{\boldsymbol{c} \sim \mathcal{C}, \boldsymbol{x}_{0:T} \sim p_{\boldsymbol{\theta}}(\boldsymbol{x}_{0:T} | \boldsymbol{c})} \Big[ R(\boldsymbol{x}_{0:T}, \boldsymbol{c}) \Big] - \beta \mathbb{D}_{\mathrm{KL}} \Big[ p_{\boldsymbol{\theta}}(\boldsymbol{x}_0 | \boldsymbol{c}) \| p_{\boldsymbol{\theta}_{\mathrm{ref}}}(\boldsymbol{x}_0 | \boldsymbol{c}) \Big]$$

$$\geq \mathbb{E}_{\boldsymbol{c} \sim \mathcal{C}, \boldsymbol{x}_{0:T} \sim p_{\boldsymbol{\theta}}(\boldsymbol{x}_{0:T} | \boldsymbol{c})} \Big[ R(\boldsymbol{x}_{0:T}, \boldsymbol{c}) \Big] - \beta \mathbb{D}_{\mathrm{KL}} \Big[ p_{\boldsymbol{\theta}}(\boldsymbol{x}_{0:T} | \boldsymbol{c}) \| p_{\boldsymbol{\theta}_{\mathrm{ref}}}(\boldsymbol{x}_{0:T} | \boldsymbol{c}) \Big]. \tag{5}$$

Following the similar derivation as in Eq. (2), the optimal joint density for the lower bound in Eq. (5) is given by:

$$p_{\boldsymbol{\theta}}^*(\boldsymbol{x}_{0:T} | \boldsymbol{c}) = \frac{1}{Z(\boldsymbol{c})} p_{\boldsymbol{\theta}_{\mathrm{ref}}}(\boldsymbol{x}_{0:T} | \boldsymbol{c}) \exp\left( \frac{1}{\beta} R(\boldsymbol{x}_{0:T}, \boldsymbol{c}) \right), \tag{6}$$

where $Z(\boldsymbol{c}) = \int_{\boldsymbol{x}_{0:T}} p_{\boldsymbol{\theta}_{\mathrm{ref}}}(\boldsymbol{x}_{0:T} | \boldsymbol{c}) \exp\left( \frac{1}{\beta} R(\boldsymbol{x}_{0:T}, \boldsymbol{c}) \right) \mathrm{d}\boldsymbol{x}_{0:T}$ is the partition function for the joint density.

### 3.3 Max-Margin Inverse Reinforcement Learning

Human preferences are represented by the labeled data originally used to train reward models. To avoid issues of error accumulation and reward hacking [10], it is advantageous to remove the reward model from preference fine-tuning entirely. This approach aligns with inverse reinforcement

learning (IRL), which aims to learn a policy based on expert demonstrations. We apply a max-margin approach [1, 23] to train a reward model that satisfies the the inductive step condition:

$$\mathbb{E}_{\boldsymbol{c}\sim\mathcal{C},\bar{\boldsymbol{x}}_0\sim\mathcal{D}(\boldsymbol{c})}\Big[r(\bar{\boldsymbol{x}}_0,\boldsymbol{c})\Big] \geq \mathbb{E}_{\boldsymbol{c}\sim\mathcal{C},\boldsymbol{x}_0\sim p_{\boldsymbol{\theta}}(\boldsymbol{x}_0|\boldsymbol{c})}\Big[r(\boldsymbol{x}_0,\boldsymbol{c})\Big], \tag{7}$$

where $\bar{\boldsymbol{x}}_0$ and $\boldsymbol{x}_0$ are samples from the expert demonstration $\mathcal{D}(\boldsymbol{c})$ and the policy model $p_{\boldsymbol{\theta}}(\boldsymbol{x}_0|\boldsymbol{c})$, respectively. Once we establish a reward model $\hat{r}(\boldsymbol{x}_0,\boldsymbol{c})$ based on Eq. (7), a new policy model $\hat{p}_{\boldsymbol{\theta}}$ is then obtained by maximizing the KL-regularized objective (Eq. (5)) using the reward model $\hat{r}(\boldsymbol{x}_0,\boldsymbol{c})$. Through altering these two optimization processes, we can obtain the optimal policy model that aligns with the expert demonstrations [1].

However, the reward model aims to maximize the margin between the expert and the policy, while the policy model seeks to minimize this margin to align more closely with the expert. This minimax optimization usually suffers from instability and an exhaustive tuning process. To this end, we further simplify the optimization procedure. First, the inductive criteria Eq. (7) can be rewritten by substituting the reward function from Eq. (4) into it:

$$\mathbb{E}_{\boldsymbol{c}\sim\mathcal{C},\bar{\boldsymbol{x}}_{0:T}\sim\mathcal{D}(\boldsymbol{c})}\Big[R(\bar{\boldsymbol{x}}_{0:T},\boldsymbol{c})\Big] \geq \mathbb{E}_{\boldsymbol{c}\sim\mathcal{C},\boldsymbol{x}_{0:T}\sim p_{\boldsymbol{\theta}}(\boldsymbol{x}_{0:T}|\boldsymbol{c})}\Big[R(\boldsymbol{x}_{0:T},\boldsymbol{c})\Big]. \tag{8}$$

We propose to parameterize the trajectory reward model $R$ by using a formulation similar to Eq. (3):

$$R_{\boldsymbol{\phi}}(\boldsymbol{x}_{0:T},\boldsymbol{c}) = \beta\log\frac{p_{\boldsymbol{\phi}}(\boldsymbol{x}_{0:T}|\boldsymbol{c})}{p_{\boldsymbol{\theta}_{\text{ref}}}(\boldsymbol{x}_{0:T}|\boldsymbol{c})} + \beta\log Z(\boldsymbol{c}), \tag{9}$$

where $\boldsymbol{\phi}$ represents the learnable parameters of the probability model $p_{\boldsymbol{\phi}}$. Assuming that there exist a reward model $\hat{R}_{\boldsymbol{\phi}}$, parameterized as in Eq. (9) and satisfying the inductive criteria in Eq. (8), we can further obtain the optimal policy by substituting $\hat{R}_{\boldsymbol{\phi}}$ into Eq. (6):

$$\begin{aligned}
\hat{p}_{\boldsymbol{\theta}}(\boldsymbol{x}_{0:T}|\boldsymbol{c}) &= \frac{p_{\boldsymbol{\theta}_{\text{ref}}}(\boldsymbol{x}_{0:T}|\boldsymbol{c})}{Z(\boldsymbol{c})}\exp\left(\frac{1}{\beta}\hat{R}_{\boldsymbol{\phi}}(\boldsymbol{x}_{0:T},\boldsymbol{c})\right) \\
&= \frac{p_{\boldsymbol{\theta}_{\text{ref}}}(\boldsymbol{x}_{0:T}|\boldsymbol{c})}{Z(\boldsymbol{c})}\exp\left(\log\frac{\hat{p}_{\boldsymbol{\phi}}(\boldsymbol{x}_{0:T}|\boldsymbol{c})}{p_{\boldsymbol{\theta}_{\text{ref}}}(\boldsymbol{x}_{0:T}|\boldsymbol{c})} + \log Z(\boldsymbol{c})\right) \\
&= \hat{p}_{\boldsymbol{\phi}}(\boldsymbol{x}_{0:T}|\boldsymbol{c}).
\end{aligned} \tag{10}$$

This result implies that the optimal policy model $\hat{p}_{\boldsymbol{\theta}}$ is identical to reward probability model $\hat{p}_{\boldsymbol{\phi}}$. Therefore, the alternating optimization reduces to reward modeling alone, where the maximum expected reward is implicitly achieved by Eq. (10) for any given $\hat{R}_{\boldsymbol{\phi}}$.

To optimize the reward model, we subtract the right hand side from the left hand side of Eq. (8), and substitute the reward parameterization from Eq. (9) into it. Moreover, we use the forward diffusion $q(\bar{\boldsymbol{x}}_{1:T}|\bar{\boldsymbol{x}}_0)$ to approximate sampling expert trajectory $\bar{\boldsymbol{x}}_{0:T}$ from the expert demonstration $\mathcal{D}(\boldsymbol{c})$:

$$\mathbb{E}_{\boldsymbol{c}\sim\mathcal{C},\bar{\boldsymbol{x}}_0\sim\mathcal{D}(\boldsymbol{c})\bar{\boldsymbol{x}}_{1:T}\sim q(\bar{\boldsymbol{x}}_{1:T}|\bar{\boldsymbol{x}}_0)}\left[\beta\log\frac{p_{\boldsymbol{\phi}}(\bar{\boldsymbol{x}}_{0:T}|\boldsymbol{c})}{p_{\boldsymbol{\theta}_{\text{ref}}}(\bar{\boldsymbol{x}}_{0:T}|\boldsymbol{c})}\right] - \mathbb{E}_{\boldsymbol{c}\sim\mathcal{C},\boldsymbol{x}_{0:T}\sim p_{\boldsymbol{\theta}}(\boldsymbol{x}_{0:T}|\boldsymbol{c})}\left[\beta\log\frac{p_{\boldsymbol{\phi}}(\boldsymbol{x}_{0:T}|\boldsymbol{c})}{p_{\boldsymbol{\theta}_{\text{ref}}}(\boldsymbol{x}_{0:T}|\boldsymbol{c})}\right].$$
$$\tag{11}$$

Through algebraic manipulation, this ranking objective is equivalent to (a detailed step-by-step derivation is provided in Appendix D):

$$\begin{aligned}
\sum_t^T \mathbb{E}_{\boldsymbol{c},\bar{\boldsymbol{x}}_0,\bar{\boldsymbol{\epsilon}}}&\left[\left\|\bar{\boldsymbol{\epsilon}} - \boldsymbol{\epsilon}_{\boldsymbol{\theta}_{\text{ref}}}(\bar{\boldsymbol{x}}_t,\boldsymbol{c},t)\right\|^2 - \left\|\bar{\boldsymbol{\epsilon}} - \boldsymbol{\epsilon}_{\boldsymbol{\phi}}(\bar{\boldsymbol{x}}_t,\boldsymbol{c},t)\right\|^2\right] \\
&- \sum_t^T \mathbb{E}_{\boldsymbol{c},\boldsymbol{x}_t}\left[\left\|\boldsymbol{\epsilon} - \boldsymbol{\epsilon}_{\boldsymbol{\theta}_{\text{ref}}}(\boldsymbol{x}_t,\boldsymbol{c},t)\right\|^2 - \left\|\boldsymbol{\epsilon} - \boldsymbol{\epsilon}_{\boldsymbol{\phi}}(\boldsymbol{x}_t,\boldsymbol{c},t)\right\|^2\right],
\end{aligned} \tag{12}$$

where $\bar{\boldsymbol{x}}_t \sim q(\bar{\boldsymbol{x}}_t|\bar{\boldsymbol{x}}_0)$ represents samples drawn from forward diffusion with perturbation noise $\bar{\boldsymbol{\epsilon}} \sim \mathcal{N}(\boldsymbol{0},\boldsymbol{I})$, and $\boldsymbol{\epsilon} = \boldsymbol{\epsilon}_{\boldsymbol{\theta}}(\boldsymbol{x}_t,\boldsymbol{c},t)$ is the noise predicted by the policy diffusion model.

**Connection to Supervised Fine-Tuning**

The ranking objective in Eq. (12) can be solved by directly minimizing the negative of the margin:

$$\mathcal{L}_{\text{mm}}(\boldsymbol{\phi}) = \sum_t^T \mathbb{E}_{\boldsymbol{c},\bar{\boldsymbol{x}}_0,\bar{\boldsymbol{\epsilon}},\boldsymbol{x}_t}\left[\underbrace{\left\|\bar{\boldsymbol{\epsilon}} - \boldsymbol{\epsilon}_{\boldsymbol{\phi}}(\bar{\boldsymbol{x}}_t,\boldsymbol{c},t)\right\|^2}_{\text{Same as SFT}} - \underbrace{\left\|\boldsymbol{\epsilon}_{\boldsymbol{\theta}}(\boldsymbol{x}_t,\boldsymbol{c},t) - \boldsymbol{\epsilon}_{\boldsymbol{\phi}}(\boldsymbol{x}_t,\boldsymbol{c},t)\right\|^2}_{\text{Push away } p_{\boldsymbol{\theta}}}\right]. \tag{13}$$

---

**Algorithm 1** Diffusion Denoising Ranking Optimization

---

**Input:** Reference diffusion model $p_{\boldsymbol{\theta}_{\mathrm{ref}}}$, prompt set $\mathcal{C}$, expert demonstration set $\{\mathcal{D}(\boldsymbol{c})\}_{\boldsymbol{c}\in\mathcal{C}}$, number of update steps $N$, policy model update interval $M$, batch size $B$, and clipping threshold $m$.

1:    $p_{\boldsymbol{\theta}} \leftarrow p_{\boldsymbol{\theta}_{\mathrm{ref}}}$                                                         ◁ Initialize policy model
2:    $p_{\boldsymbol{\phi}} \leftarrow p_{\boldsymbol{\theta}_{\mathrm{ref}}}$                                                         ◁ Initialize reward model
3: **for** $i = 1$ to $N$ **do**
4:      **for** $n = 1$ to $B$ **do**
5:          $t \sim \mathcal{U}\{1, T\}$
6:          $\boldsymbol{c} \overset{iid}{\sim} \mathcal{C}, \bar{\boldsymbol{x}}_0 \overset{iid}{\sim} \mathcal{D}(\boldsymbol{c}), \bar{\boldsymbol{\epsilon}} \sim \mathcal{N}(\mathbf{0}, \boldsymbol{I})$
7:          $\bar{\boldsymbol{x}}_t \sim q(\bar{\boldsymbol{x}}_t | \bar{\boldsymbol{x}}_0)$                                     ◁ Forward diffusion
8:          $\boldsymbol{x}_t \sim p_{\boldsymbol{\theta}}(\boldsymbol{x}_t | \boldsymbol{c})$                                  ◁ Sample from policy
9:          $\boldsymbol{\epsilon} \leftarrow \boldsymbol{\epsilon}_{\boldsymbol{\theta}}(\boldsymbol{x}_t, \boldsymbol{c}, t)$
10:        $\mathcal{L}_{\mathrm{L}}^n \leftarrow \left\| \bar{\boldsymbol{\epsilon}} - \boldsymbol{\epsilon}_{\boldsymbol{\theta}_{\mathrm{ref}}}(\bar{\boldsymbol{x}}_t, \boldsymbol{c}, t) \right\|^2 - \left\| \bar{\boldsymbol{\epsilon}} - \boldsymbol{\epsilon}_{\boldsymbol{\phi}}(\bar{\boldsymbol{x}}_t, \boldsymbol{c}, t) \right\|^2$
11:        $\mathcal{L}_{\mathrm{R}}^n \leftarrow \left\| \boldsymbol{\epsilon} - \boldsymbol{\epsilon}_{\boldsymbol{\theta}_{\mathrm{ref}}}(\boldsymbol{x}_t, \boldsymbol{c}, t) \right\|^2 - \left\| \boldsymbol{\epsilon} - \boldsymbol{\epsilon}_{\boldsymbol{\phi}}(\boldsymbol{x}_t, \boldsymbol{c}, t) \right\|^2$
12:      **end for**
13:      $\mathcal{L}_{\mathrm{TRL}}(\boldsymbol{\phi}) \leftarrow \frac{1}{B} \sum_{n=1}^{B} \max\left(m, -\mathcal{L}_{\mathrm{L}}^n + \mathcal{L}_{\mathrm{R}}^n\right)$                           ◁ Eq. (15)
14:      Update $\boldsymbol{\phi}$ using gradient $\nabla_{\boldsymbol{\phi}} \mathcal{L}_{\mathrm{TRL}}(\boldsymbol{\phi})$
15:      **if** $i$ is multiple of $M$ **then**
16:          $p_{\boldsymbol{\theta}} \leftarrow p_{\boldsymbol{\phi}}$                                        ◁ Update policy model
17:      **end if**
18: **end for**

---

We eliminate terms that do not depend on $\phi$, as they do not contribute to the gradients of reward model. We notice that supervised fine-tuning corresponds to optimizing the first term, which minimizes the KL-divergence between the expert distribution and the distribution induced by the reward model. The second term serves a complementary purpose, where the reward model generates negative samples online to guide the optimization in the correct direction.

**Connection to DPO-based Approaches**

Previous DPO-based approaches [35, 39, 30, 40] aim to learn preference predictions using the Bradley-Terry [4] model. Diffusion-DPO applies Jensen's inequality to transform the objective from a probability-based form to a noise-prediction form (Eq.(14) in Wallace et al. 35). This transformation closely resembles solving the ranking problem in Eq. (12) by maximizing the margin using a cross-entropy loss:

$$
\mathcal{L}_{\mathrm{ce}}(\boldsymbol{\phi}) = -\log \sigma \left( \sum_t^T \mathbb{E}_{\boldsymbol{c}, \bar{\boldsymbol{x}}_0, \bar{\boldsymbol{\epsilon}}, \boldsymbol{x}_t} \left[ \left( \left\| \bar{\boldsymbol{\epsilon}} - \boldsymbol{\epsilon}_{\boldsymbol{\theta}_{\mathrm{ref}}}(\bar{\boldsymbol{x}}_t, \boldsymbol{c}, t) \right\|^2 - \left\| \bar{\boldsymbol{\epsilon}} - \boldsymbol{\epsilon}_{\boldsymbol{\phi}}(\bar{\boldsymbol{x}}_t, \boldsymbol{c}, t) \right\|^2 \right) \right. \right.
$$
$$
\left. \left. - \left( \left\| \boldsymbol{\epsilon} - \boldsymbol{\epsilon}_{\boldsymbol{\theta}_{\mathrm{ref}}}(\boldsymbol{x}_t, \boldsymbol{c}, t) \right\|^2 - \left\| \boldsymbol{\epsilon} - \boldsymbol{\epsilon}_{\boldsymbol{\phi}}(\boldsymbol{x}_t, \boldsymbol{c}, t) \right\|^2 \right) \right] \right),
\tag{14}
$$

where $\sigma(x) = 1/(1 + \exp(-x))$ denotes the sigmoid function. Unlike prior work, our approach does not require Jensen's inequality to minimize the surrogate upper bound. The max-margin approach enables direct optimization of the margin while ensuring convergence of the policy model. Specifically, we sample pairs from expert demonstration and policy density, whereas DPO-based methods compare preference pairs $(\boldsymbol{x}^w, \boldsymbol{x}^l)$, where $\boldsymbol{x}^w$ is preferred over $\boldsymbol{x}^l$. In other words, the proposed inverse RL approach decouples the need for preference pairs and further eliminates the reliance on negative samples. In practice, public preferences can be obtained through simple statistical methods to rank different samples, with higher-ranked ones treated as expert demonstrations that align with public preferences. The same method can be easily extended to collect expert demonstrations reflecting individual preferences.

### 3.4 Thresholded Ranking Loss

While our proposed reward parameterization reduces the learnable parameters to include only the reward model, two separate models are still maintained to represent the reward and policy, respectively.

To ensure stable reward model updates, the policy model is periodically synchronized with the latest reward model after a predefined number of gradient steps (see line 16 in Alg. 1). However, updating the policy too frequently can limit the reward model's ability to adapt and distinguish expert behaviors from policy outputs. Conversely, infrequent updates may cause the reward model to overfit to the current policy. To balance this trade-off, we introduce a thresholded ranking loss (TRL), which clips the margin loss once the inductive criterion is sufficiently satisfied:

$$
\mathcal{L}_{\text{TRL}}(\boldsymbol{\phi}) = \sum_t^T \mathbb{E}_{\boldsymbol{c}, \bar{\boldsymbol{x}}_0, \bar{\boldsymbol{\epsilon}}, \boldsymbol{x}_t} \left[ \max \left( m, - \left( \left\| \bar{\boldsymbol{\epsilon}} - \boldsymbol{\epsilon}_{\boldsymbol{\theta}_{\text{ref}}}(\bar{\boldsymbol{x}}_t, \boldsymbol{c}, t) \right\|^2 - \left\| \bar{\boldsymbol{\epsilon}} - \boldsymbol{\epsilon}_{\boldsymbol{\phi}}(\bar{\boldsymbol{x}}_t, \boldsymbol{c}, t) \right\|^2 \right) \right. \right.
$$
$$
\left. \left. + \left( \left\| \boldsymbol{\epsilon} - \boldsymbol{\epsilon}_{\boldsymbol{\theta}_{\text{ref}}}(\boldsymbol{x}_t, \boldsymbol{c}, t) \right\|^2 - \left\| \boldsymbol{\epsilon} - \boldsymbol{\epsilon}_{\boldsymbol{\phi}}(\boldsymbol{x}_t, \boldsymbol{c}, t) \right\|^2 \right) \right) \right], \tag{15}
$$

where $m$ is a predefined parameter that adjusts the baseline at which the reward margin is truncated. For samples that already satisfy the inductive criterion, further optimization of the margin is unnecessary. By clipping the margin in these cases, the model can concentrate on rectifying incorrect rankings, thereby avoiding overfitting to samples that are already ranked correctly.

We refer to the learning process with the objective $\mathcal{L}_{\text{TRL}}(\boldsymbol{\phi})$ as Diffusion Denoising Ranking Optimization (Diffusion-DRO). This method learns the ranking relationships between expert demonstrations and policy behaviors. The training process is detailed in Algorithm 1.

## 4 Experiments

We first outline the datasets, implementation details, and evaluation protocols used in our experiments. We then evaluate Diffusion-DRO against multiple baselines using quantitative metrics and supplement our findings with a user study on Amazon Mechanical Turk for qualitative comparison.

**Datasets**. Following prior works [17, 35, 18], we use the train split of Pick-a-Pic v2 [14] (MIT license) as our training dataset. For evaluation, we adopt the test split of Pick-a-Pic v2 and the HPDv2 benchmark [36] (Apache-2.0 license), representing in-domain and out-of-domain scenarios, respectively. Each sample includes a prompt, two images, and a human preference label. Due to label sparsity, we simulate expert demonstrations using automated metrics such as PickScore [14] (MIT license) and HPSv2 [36] (Apache-2.0 license). We rank all training pairs by these scores and select the top $K$ as expert demonstrations; unless otherwise stated, $K=500$. Ablation results for varying $K$ are provided in Section 4.4.

**Evaluation**. We evaluate the human preference alignment by comparing it with various baseline methods. Preference scores are computed using five different score models: PickScore [14], HPSv2 [36], Aesthetic [34] (MIT license), CLIP Score [29] (MIT license) and ImageReward [38] (Apache-2.0 license). For each preference score, we report the win rates between the Diffusion-DRO and the baseline methods, defined as the proportion of generated results with scores exceeding those of the baselines. To ensure fairness, we avoid using the same preference score for selecting the expert demonstrations and calculating the win rates, as this could inadvertently leak score prior information into the train data. Specifically, we use HPSv2 to select the expert demonstrations and calculate win rates for all metrics except for HPSv2. The experiments using different metrics to select expert demonstrations can be found in Appendix B.

**Implementation Details**. We fine-tune Stable Diffusion 1.5 (SD v1-5) [31] (CreativeML Open RAIL-M license) using Diffusion-DRO, ensuring consistency across all baseline methods. To sample $\boldsymbol{x}_t$ from the policy model, we employ DPMSolver++ [21] with 20 steps, without using classifier-free guidance [12]. For inference, we use the DDPM sampler with 50 steps and a classifier-free guidance scale of 7.5 to generate five images per prompt for all methods. Among these five generations, we select the image with the median PickScore as the final result. For additional implementation details, please refer to Appendix A.2.

### 4.1 Quantitative Results

We compare Diffusion-DRO with strong baselines, including SPIN-Diffusion [41] (Apache-2.0 license), Diffusion-SPO [18] (Apache-2.0 license), Diffusion-DPO [35] (Apache-2.0 license), and

Table 1: Automated win rates between Diffusion-DRO and baseline methods. The dagger symbol ($\dagger$) indicates that the evaluation was performed on the officially released model weights. Note that SD v1-5 w/ SFT refers to SD v1-5 fine-tuned on expert demonstrations. Win rates greater than 50% are highlighted in bold.

| Baseline Method | Pick-a-Pic v2 Test | | | | HPDv2 Benchmark | | | |
|---|---|---|---|---|---|---|---|---|
| | PickScore | Aesthetic Score | CLIP Score | ImageReward | PickScore | Aesthetic Score | CLIP Score | ImageReward |
| SD v1-5 [†] | **87.80** | **85.20** | 48.40 | **88.60** | **90.47** | **82.91** | 46.59 | **87.69** |
| SD v1-5 w/ SFT | **71.20** | **58.00** | **66.40** | **57.80** | **70.62** | **57.22** | **64.97** | **62.03** |
| SPIN-Diffusion [†] | **56.20** | **64.80** | **58.20** | **70.60** | **54.87** | **62.78** | **54.78** | **69.78** |
| Diffusion-SPO [†] | **62.80** | **63.60** | **71.40** | **78.00** | **60.59** | **67.66** | **75.78** | **77.94** |
| Diffusion-SPO w/ SFT | **86.60** | **81.60** | 42.40 | **87.20** | **88.75** | **80.25** | 42.69 | **85.78** |
| Diffusion-DPO [†] | **78.40** | **83.20** | 41.40 | **84.20** | **79.75** | **79.97** | 39.09 | **82.25** |
| Diffusion-DPO w/ SFT | **64.00** | **55.00** | **59.00** | **56.20** | **63.62** | **56.12** | **59.91** | **58.75** |
| Diffusion-KTO [†] | **74.20** | **69.00** | 42.20 | **66.60** | **71.19** | **71.03** | 39.81 | **62.81** |
| Diffusion-KTO w/ SFT | **70.20** | **58.60** | **64.00** | **58.60** | **71.09** | **56.12** | **65.31** | **62.75** |

Diffusion-KTO [17]. These methods remove the need for a reward model and are fine-tuned from SD v1-5, consistent with our setup.

Since both expert selection and evaluation rely on automated metrics, there may be concerns about potential information leakage. To address this, we additionally fine-tune SD v1-5 on our selected expert demonstrations and select the best checkpoint based on PickScore performance on the Pick-a-Pic v2 test set, denoting it as SD v1-5 w/ SFT. Using this model, we further fine-tune Diffusion-DPO, Diffusion-KTO, and Diffusion-SPO with their official implementations[1]. These variants are labeled with the postfix "w/ SFT." Table 1 reports the win rates of Diffusion-DRO against all baselines using various automated preference scores. Full results with raw scores and standard deviations are provided in Appendix B.

For the SD v1-5 w/ SFT, performance improves in PickScore, Aesthetic, and ImageReward compared to SD v1-5 (resulting in lower win rates for our method). We attribute this to expert demonstrations enhancing preference alignment. However, an interesting observation is the decline in CLIP Score. This can be attributed to the model slightly deviating from the original text encoder distribution, which was trained on large-scale data. Since Stable Diffusion and CLIP use identical text encoder weights, this deviation leads to a decrease in CLIP Score for SD v1-5 w/ SFT. The same phenomenon is also observed in Diffusion-DPO w/ SFT and Diffusion-KTO w/ SFT since they are fine-tuned from SD v1-5 w/ SFT. For the Diffusion-SPO w/ SFT, their proposed step-aware preference model leverages the CLIP vision and text encoders to select the best and worst samples for fine-tuning. Therefore, the CLIP Score of Diffusion-SPO w/ SFT increases due to the consistent distribution.

We observe that Diffusion-DRO significantly outperforms all state-of-the-art approaches across multiple metrics, including PickScore, Aesthetic, and ImageReward. Even when compared to stronger baselines, such as Diffusion-KTO w/ SFT and Diffusion-DPO w/ SFT, Diffusion-DRO remains the preferred method in terms of all automated evaluation scores. For Diffusion-SPO w/ SFT, its reliance on online sampling for step-aware preference pairs makes it susceptible to the generation quality and diversity of the pre-trained Stable Diffusion model. While fine-tuning Stable Diffusion with expert demonstrations enhances preference alignment, it also reduces the variation in step-aware preference pairs. As a result, Diffusion-SPO w/ SFT fails to gain any performance improvement over SD v1-5 w/ SFT. Consequently, Diffusion-DRO significantly outperforms Diffusion-SPO w/ SFT, achieving win rates exceeding 80% across PickScore, Aesthetic, and ImageReward.

Notably, Diffusion-DRO is fine-tuned directly from SD v1-5, unlike strong baseline methods such as Diffusion-DPO w/ SFT and Diffusion-KTO w/ SFT. Despite this, Diffusion-DRO outperforms methods that start fine-tuning from SD v1-5 w/ SFT, demonstrating its capacity to effectively learn human preferences from expert demonstrations.

---

[1]SPIN-Diffusion [41] does not provide valid source code for fine-tuning from SD v1-5 w/ SFT.

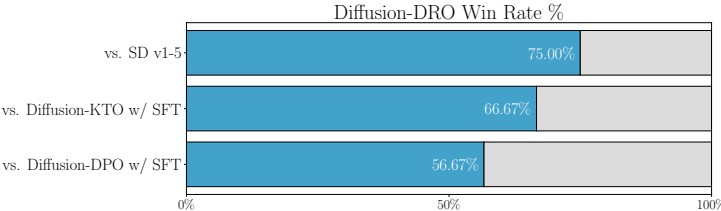

Figure 1: User study results comparing Diffusion-DRO with baseline methods. The win rate represents the proportion of survey questions where users preferred Diffusion-DRO over the baselines.

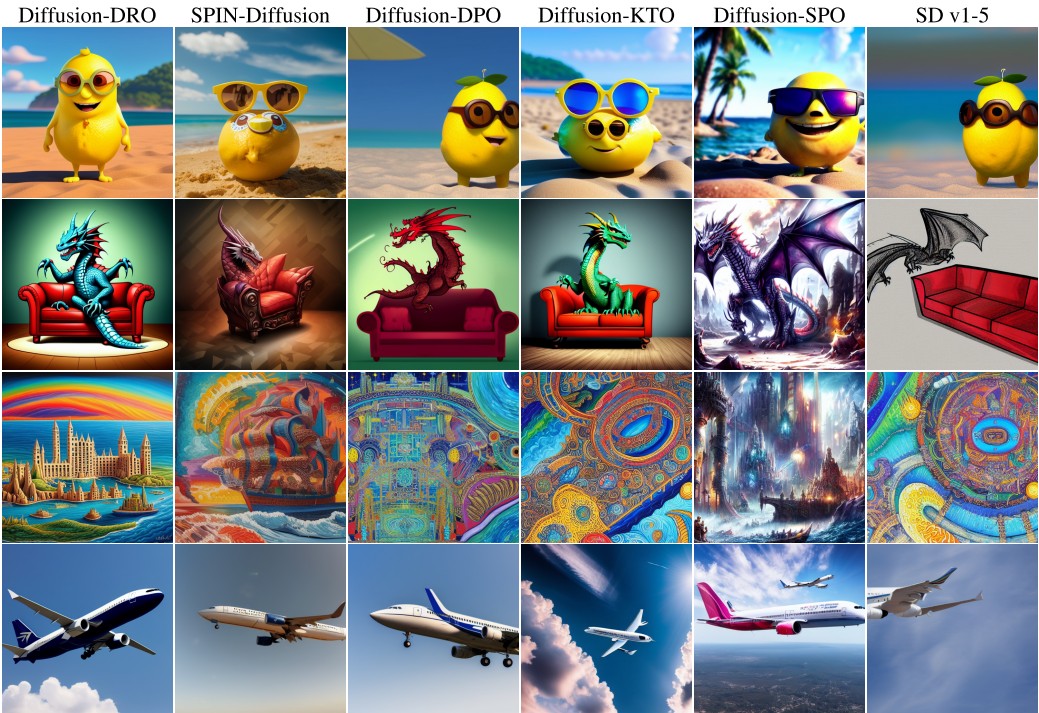

Figure 2: From top to bottom, the text prompts are: "*A Pixar lemon wearing sunglasses on a beach,*" "*A dragon sitting on a couch in a digital illustration,*" "*A detailed painting of Atlantis by multiple artists, featuring intricate detailing and vibrant colors,*" and "*A passenger jet aircraft flying in the sky.*"

Compared to SPIN-Diffusion, both methods use generations from diffusion models as negative samples. However, Diffusion-DRO simplifies the training process into a single stage by adopting a max-margin inverse reinforcement learning (IRL) formulation. This advantage allows Diffusion-DRO to achieve over a 60% win rate on Aesthetic Score and nearly a 70% win rate on ImageReward, outperforming SPIN-Diffusion.

## 4.2 User Study

The user study compares Diffusion-DRO with baseline methods, including SD v1-5, Diffusion-DPO w/ SFT, and Diffusion-KTO w/ SFT. Text prompts are randomly sampled from the HPDv2 Benchmark to generate images for evaluation. Detailed settings of the user study are provided in Appendix C.

Figure 1 presents the results of our user study, showing that Diffusion-DRO achieves a 75% win rate against SD v1-5. This demonstrates the effectiveness of our training procedure in improving the pre-trained SD model, as human evaluators consistently prefer images generated by Diffusion-DRO. Additionally, the win rates of Diffusion-DRO against Diffusion-DPO (56.67%) and Diffusion-KTO (66.67%) further support the reliability of Table 1. For instance, the average win rate against Diffusion-DPO w/ SFT is 59.6%, and against Diffusion-KTO w/ SFT is 63.82%, closely aligning with the user study findings.

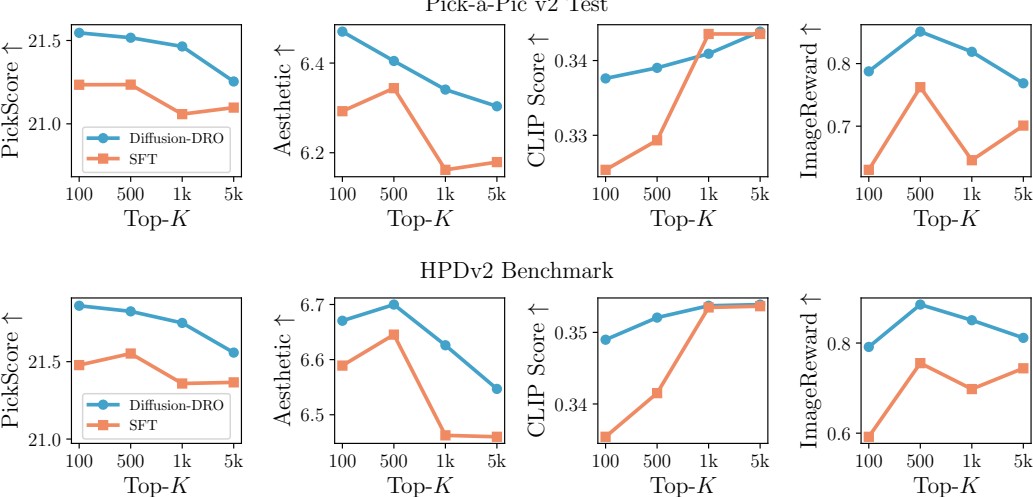

Figure 3: Evaluation results of Diffusion-DRO and SFT trained with varying amounts of expert demonstrations.

## 4.3 Qualitative Results

In Figure 2, we present the generation results of different methods. The example prompts from top to bottom are sampled from the four categories of the HPDv2 Benchmark, namely Anime, Concept Art, Paintings, and Photos. In the first row, Diffusion-DRO successfully generates a "lemon wearing sunglasses," while SPIN-Diffusion, Diffusion-KTO, and SD v1-5 fail to produce accurate results. In the second row, both Diffusion-DRO and Diffusion-KTO correctly generate the "dragon," the couch," and the action "sit," whereas other methods produce incorrect objects or actions. For the third row, Diffusion-DRO captures the intricate details of Atlantis, while Diffusion-DPO and Diffusion-KTO generate abstract content. In the final row, only Diffusion-DRO produces a realistic airplane, whereas the outputs from other methods result in implausible shapes. These examples highlight that Diffusion-DRO significantly improves both text alignment and visual fidelity.

## 4.4 Ablation of Expert Demonstration

In previous experiments, we observe that SFT delivers competitive performances. Therefore, we are interested in exploring the performance disparity between Diffusion-DRO and SFT under different volumes of training data. To investigate this further, we utilize HPSv2 to select varying quantities of expert demonstrations. These demonstrations are then used to train both Diffusion-DRO and SFT models. The results, depicted in Figure 3, reveal that the CLIP Score consistently increases with the size of the training dataset. This phenomenon can be attributed to the fact that the text encoder in SD v1-5 is identical to the one used in CLIP, resulting in improved scores as the training data volume increases.

Furthermore, across various test sets, Diffusion-DRO outperforms SFT in terms of PickScore, Aesthetic, and ImageReward metrics. These results demonstrate that the thresholded ranking loss consistently enhances Diffusion-DRO's alignment with human preferences. We attribute this improvement to a fundamental difference in learning objectives. That is, SFT focuses solely on minimizing KL divergence, which neglects the additional expert priors embedded in expert demonstrations. In contrast, Diffusion-DRO leverages these priors by treating policy actions as negative samples, thereby enabling more effective training.

## 5 Conclusion

We propose Diffusion-DRO, a preference learning framework for text-to-image diffusion models based on inverse reinforcement learning. By reformulating the objective to remove the non-linear sigmoid function, our method simplifies optimization into a denoising task, improving training efficiency and stability. Diffusion-DRO further balances offline and online training by combining expert

demonstrations with policy-generated negatives, addressing the limitations of offline data. Comprehensive experimental evaluations and user studies demonstrate that Diffusion-DRO consistently outperforms state-of-the-art baseline methods across diverse and unseen prompts. By integrating human preferences more effectively, our method achieves superior generation quality, making it a robust and scalable solution for preference alignment in text-to-image generation tasks.

## Acknowledgments

This work is partially supported by the National Science and Technology Council, Taiwan, under Grant: NSTC-112-2221-E-A49-059-MY3 and NSTC-112-2221-E-A49-094-MY3.

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

# A    Experiment Details

## A.1    Datasets

We use the Pick-a-Pic v2 [14] training set as the source of expert demonstrations, consisting of 959,040 preference pairs and 1,029,802 unique text-image pairs. Note that the train set of Diffusion-SPO [18] is Pick-a-Pic v1, which is a little different from our settings, but in a comparable range. Our train set is consistent with other baseline methods, i.e., Diffusion-DPO [35] (Apache-2.0 license) and Diffusion-KTO [17]. For testing, we utilize two datasets to ensure diverse evaluation scenarios. The first is the Pick-a-Pic v2 test set, which includes 500 unique text prompts collected from users of the deployed web application. The second is the HPSv2 Benchmark, divided into four categories: anime, concept art, painting, and photo. Each category contains 800 text prompts. However, we observe slight discrepancies in the number of unique prompts: 781 for anime, 795 for concept art, 798 for painting, and 800 for photo. To maintain consistency with prior works [17, 8], we use all 800 prompts (including duplicates) for each category during testing. To select the expert demonstrations from Pick-a-Pic v2, we use a preference metric to give each text-image pair a score representing the quality of being an expert demonstration. We then sort all pairs in descending order by the scores and select the top $K$ pairs as the expert demonstrations. If not otherwise specified, $K = 500$ is used.

## A.2    Implementation Details

We fine-tune Stable Diffusion 1.5 (SD v1-5) [31] using Diffusion-DRO. The AdamW optimizer [20] is used with a learning rate of $10^{-4}$ and an effective batch size of 256 (4 samples per GPU, 32 gradient accumulation steps, yielding $4 \times 4 \times 16 = 256$). The training consists of 1,600 optimization steps, resulting in a total of $16 \times 1,600 = 25,600$ iterations when accounting for gradient accumulation. During training, 20% of prompts are randomly replaced with empty strings, which helps preserve the model's ability to perform unconditional generation by maintaining a balance between conditional and unconditional sampling.

Following the standard Stable Diffusion training process, we apply an exponential moving average (EMA) to aggregate the UNet weights during training, with a decay rate of 0.9999. The clipping threshold $m$ for the thresholded ranking loss (TRL) is set to $-0.001$, and the policy model update interval $M$ is set to 1 for all experiments.

For sampling $x_t$ from the policy model, we employ DPMSolver++ [21] with 20 steps, without utilizing classifier-free guidance [12]. To ensure all time steps in SD v1-5 are adequately fine-tuned, we uniformly perturb the sampling time steps of DPMSolver++ during training. This approach allows the time steps used during inference to differ from those used for online sampling in training, enhancing the model's robustness across all time steps.

All experiments, including the reproduction of baseline methods with updated SD model weights, were conducted on four NVIDIA RTX 3090 GPUs. Training Diffusion-DRO takes approximately 20 to 25 hours.

Table 2: Automated win rates between Diffusion-DRO and baseline methods. PickScore is used to select expert demonstrations. The dagger symbol ([†]) indicates that the evaluation was performed on the officially released model weights. Note that SD v1-5 w/ SFT refers to SD v1-5 fine-tuned on expert demonstrations. Win rates greater than 50% are highlighted in bold.

| Baseline Method | Pick-a-Pic v2 Test | | | | HPDv2 Benchmark | | | |
|---|---|---|---|---|---|---|---|---|
| | HPSv2 | Aesthetic Score | CLIP Score | ImageReward | PickScore | Aesthetic Score | CLIP Score | ImageReward |
| SD v1-5 [†] | **94.80** | **79.00** | **58.60** | **87.40** | **97.00** | **79.44** | **51.06** | **88.59** |
| SD v1-5 w/ SFT | **71.40** | **54.40** | **68.80** | **60.60** | **70.34** | **55.84** | **69.06** | **61.53** |
| SPIN-Diffusion [†] | **78.20** | **57.60** | **65.80** | **72.40** | **80.56** | **59.81** | **61.78** | **72.59** |
| Diffusion-SPO [†] | **84.40** | **57.60** | **76.80** | **78.80** | **85.78** | **65.06** | **79.19** | **79.16** |
| Diffusion-DPO [†] | **92.20** | **79.00** | **51.80** | **83.00** | **94.31** | **77.28** | 44.81 | **84.00** |
| Diffusion-KTO [†] | **77.40** | **68.60** | **53.60** | **66.20** | **74.25** | **66.22** | 46.91 | **65.22** |

Table 3: Preference scores of Diffusion-DRO and baseline methods evaluated on the Pick-a-Pic v2 test set. The metric used to select expert demonstrations for Diffusion-DRO is indicated in parentheses after "Diffusion-DRO", e.g., "Diffusion-DRO (PickScore)." Moreover, baseline methods with the suffix "w/ SFT" are fine-tuned from "SD v1-5 w/ SFT", which itself is fine-tuned from SD v1-5 using expert demonstrations selected by HPSv2.

| Method | PickScore | HPSv2 | Aesthetic | CLIP Score | ImageReward |
|---|---|---|---|---|---|
| SD v1-5 [†] | 20.68±1.36 | 26.88±1.81 | 5.93±1.05 | 0.3369±.058 | 0.1765±1.07 |
| SD v1-5 w/ SFT | 21.24±1.41 | 28.11±1.73 | 6.34±.974 | 0.3293±.059 | 0.7623±.906 |
| SPIN-Diffusion [†] | 21.40±1.38 | 27.78±1.79 | 6.26±.985 | 0.3305±.059 | 0.5619±.971 |
| Diffusion-SPO [†] | 21.17±1.41 | 27.35±1.75 | 6.23±.981 | 0.3150±.060 | 0.3278±1.07 |
| Diffusion-SPO w/ SFT | 20.75±1.37 | 26.93±1.75 | 5.93±1.07 | 0.3419±.056 | 0.2254±1.06 |
| Diffusion-DPO [†] | 21.00±1.40 | 27.22±1.80 | 5.96±1.05 | 0.3427±.057 | 0.3369±1.05 |
| Diffusion-DPO w/ SFT | 21.31±1.40 | 28.08±1.72 | 6.35±.995 | 0.3334±.059 | 0.7912±.901 |
| Diffusion-KTO [†] | 21.17±1.36 | 27.88±1.77 | 6.20±.954 | 0.3438±.056 | 0.6743±.962 |
| Diffusion-KTO w/ SFT | 21.25±1.41 | 28.11±1.74 | 6.34±.974 | 0.3296±.059 | 0.7636±.910 |
| Diffusion-DRO (HPSv2) | 21.52±1.42 | **28.49±1.75** | **6.40±.976** | 0.3390±.057 | 0.8511±.852 |
| Diffusion-DRO (PickScore) | **21.76±1.51** | 28.38±1.78 | 6.40±.935 | **0.3446±.057** | **0.8636±.907** |

Table 4: Preference scores of Diffusion-DRO and baseline methods evaluated on HPDv2 Benchmark. The score name that is used to select the expert demonstrations for Diffusion-DRO are denoted in the parentheses after "Diffusion-DRO", e.g., "Diffusion-DRO (PickScore)." Moreover, the baseline methods with suffix "w/ SFT" are fine-tuned from "SD v1-5 w/ SFT", which is also a fine-tuned from SD v1.5 with expert demonstrations selected by HPSv2.

| Method | PickScore | HPSv2 | Aesthetic | CLIP Score | ImageReward |
|---|---|---|---|---|---|
| SD v1-5 [†] | 20.92±1.20 | 27.36±1.66 | 6.22±.923 | 0.3532±.052 | 0.2242±.976 |
| SD v1-5 w/ SFT | 21.55±1.32 | 28.74±1.61 | 6.65±.855 | 0.3415±.053 | 0.7554±.871 |
| SPIN-Diffusion [†] | 21.74±1.21 | 28.38±1.64 | 6.54±.843 | 0.3451±.054 | 0.6071±.924 |
| Diffusion-SPO [†] | 21.63±1.25 | 28.01±1.54 | 6.47±.868 | 0.3217±.054 | 0.4141±.968 |
| Diffusion-SPO w/ SFT | 20.99±1.19 | 27.39±1.64 | 6.26±.933 | 0.3556±.051 | 0.2619±.962 |
| Diffusion-DPO [†] | 21.30±1.19 | 27.75±1.67 | 6.29±.920 | **0.3584±.051** | 0.4070±.956 |
| Diffusion-DPO w/ SFT | 21.66±1.27 | 28.77±1.59 | 6.64±.826 | 0.3454±.052 | 0.8001±.861 |
| Diffusion-KTO [†] | 21.51±1.18 | 28.57±1.61 | 6.47±.852 | 0.3579±.052 | 0.7529±.871 |
| Diffusion-KTO w/ SFT | 21.56±1.32 | 28.74±1.61 | 6.64±.855 | 0.3416±.053 | 0.7557±.870 |
| Diffusion-DRO (HPSv2) | 21.82±1.29 | **29.05±1.66** | **6.70±.873** | 0.3521±.053 | 0.8853±.843 |
| Diffusion-DRO (PickScore) | **22.04±1.28** | 28.99±1.62 | 6.66±.828 | 0.3560±.053 | **0.9069±.833** |

# B  Additional Quantitative Results

To alleviate the variation of evaluation results, we sample 5 images per prompt for all models in our benchmark. Specifically, we sample 2500 images for Pick-a-Pic v2 test and 16000 images for HPDv2 Benchmark. When calculating the win rates, we sort 5 images for each prompt according to the corresponding PickScore and select the image with medium score as the comparison target.

We report the win rates of Diffusion-DRO trained with HPSv2 selected expert demonstrations in Table 1. For the Diffusion-DRO trained with PickScore selected expert demonstrations. The win rates against baseline methods are show in Table 2. Due to the limited computation resources, we do not reproduce the baseline methods based on the new SD model (SD v1-5 w/SFT in Table 2). We only compare the Diffusion-DRO with the officially released model weights.

We present the average preference scores in Table 3 and Table 4, including PickScore [14], HPSv2 [36], Aesthetic [34], CLIP Score [29], and ImageReward [38].

# C  User Study Settings

As shown in Table 1, the automated win rates of our method are decreased after Diffusion-DPO and Diffusion-KTO using the new SD model as base weights, e.g., the win rate compared to Diffusion-

DPO and Diffusion-DPO w/ SFT on HPDv2 Benchmark decreases from 79.75 to 63.62 evaluated by PickScore. This shows that Diffusion-DPO w/ SFT and Diffusion-KTO w/ SFT are more competitive than the official released models. Therefore, we choose these two baseline models plus an SD v1-5 to be baseline methods in user studies. We use the same images generated for calculating metrics in Table 3, Table 4 and Table 1. We prepare the prompts by random sampling 60 prompts from four categoryies of HPDv2 Benchmark (15 prompts for each category). For each category, we use PickScore to sort the samples for each method and select the image with medium score as the survey target. To avoid survey participants identifying our generation results, we re-sample the prompts for each user study between Diffusion-DRO and baseline methods. This could prevent our samples from repeated occurences in different user studies.

We employ human evaluators via Amazon Mechanical Turk (MTurk) for our user studies. Although the HPDv2 Benchmark includes additional filtering steps to remove inappropriate prompts, we still indicate that the user survey may contain adult content. Before beginning the survey, users must check the box labeled **"WARNING: This HIT may contain adult content. Worker discretion is advised."**

On the survey page, participants can access the evaluation guidelines, which include the following instructions:

> For each text prompt, two AI-generated images will be displayed side by side. You can evaluate which image better meets human expectations based on (but not limited to) the following criteria. The importance of each criterion depends on your subjective judgment:
>
> - Completeness of details
>
> - Artistic or aesthetic quality
>
> - Alignment between the image and the given prompt
>
> In short, select the image that you believe demonstrates better generation quality.

On each selection page, the prompt is displayed along with two images labeled **Image A** and **Image B**, accompanied by the question: **"Which image do you prefer given the prompt?"** Below the question, two radio buttons allow users to select either Image A or Image B, with at least one selection required before submission. To ensure fairness, the images generated by Diffusion-DRO and the baseline methods are randomly assigned to Image A and Image B. Additionally, their sources cannot be identified through the webpage's source code.

For each prompt, we collect 35 responses. If the majority of these responses favor Diffusion-DRO, the prompt is considered to prefer Diffusion-DRO. Finally, we compute the proportion of prompts that favor our method as the win rate, which is reported in Figure 1.

## D  Derivation of Denoising Ranking Optimization

For convenience, we repeat Eq. (11) below:

$$\mathbb{E}_{\boldsymbol{c}\sim\mathcal{C},\bar{\boldsymbol{x}}_0\sim\mathcal{D}(\boldsymbol{c}),\bar{\boldsymbol{x}}_{1:T}\sim q(\bar{\boldsymbol{x}}_{1:T}|\bar{\boldsymbol{x}}_0)}\left[\beta\log\frac{p_\phi(\bar{\boldsymbol{x}}_{0:T}|\boldsymbol{c})}{p_{\boldsymbol{\theta}_{\text{ref}}}(\bar{\boldsymbol{x}}_{0:T}|\boldsymbol{c})}\right] - \mathbb{E}_{\boldsymbol{c}\sim\mathcal{C},\boldsymbol{x}_{0:T}\sim p_{\boldsymbol{\theta}}(\boldsymbol{x}_{0:T}|\boldsymbol{c})}\left[\beta\log\frac{p_\phi(\boldsymbol{x}_{0:T}|\boldsymbol{c})}{p_{\boldsymbol{\theta}_{\text{ref}}}(\boldsymbol{x}_{0:T}|\boldsymbol{c})}\right].$$

$$(16)$$

The first term on the left-hand side (LHS) and second term on the right-hand side (RHS) share similar simplification processes. We first present the derivation of the LHS:

$$\mathbb{E}_{\boldsymbol{c}\sim\mathcal{C},\bar{\boldsymbol{x}}_0\sim\mathcal{D}(\boldsymbol{c}),\bar{\boldsymbol{x}}_{1:T}\sim q(\bar{\boldsymbol{x}}_{1:T}|\bar{\boldsymbol{x}}_0)}\left[\beta\log\frac{p_\phi(\bar{\boldsymbol{x}}_{0:T}|\boldsymbol{c})}{p_{\boldsymbol{\theta}_{\text{ref}}}(\bar{\boldsymbol{x}}_{0:T}|\boldsymbol{c})}\right] \tag{17}$$

$$=\mathbb{E}_{\boldsymbol{c}\sim\mathcal{C},\bar{\boldsymbol{x}}_0\sim\mathcal{D}(\boldsymbol{c}),\bar{\boldsymbol{x}}_{1:T}\sim q(\bar{\boldsymbol{x}}_{1:T}|\bar{\boldsymbol{x}}_0)}\left[\beta\sum_{t=1}^{T}\log\frac{p_\phi(\bar{\boldsymbol{x}}_{t-1}|\bar{\boldsymbol{x}}_t,\boldsymbol{c})}{p_{\boldsymbol{\theta}_{\text{ref}}}(\bar{\boldsymbol{x}}_{t-1}|\bar{\boldsymbol{x}}_t,\boldsymbol{c})} + \beta\log\frac{p_\phi(\boldsymbol{x}_T|\boldsymbol{c})}{p_{\boldsymbol{\theta}_{\text{ref}}}(\boldsymbol{x}_T|\boldsymbol{c})}\right] + C \tag{18}$$

$$=\beta\sum_{t=1}^{T}\mathbb{E}_{\boldsymbol{c}\sim\mathcal{C},\bar{\boldsymbol{x}}_0\sim\mathcal{D}(\boldsymbol{c}),\bar{\boldsymbol{x}}_t\sim q(\bar{\boldsymbol{x}}_t|\bar{\boldsymbol{x}}_0),\bar{\boldsymbol{x}}_{t-1}\sim q(\bar{\boldsymbol{x}}_{t-1}|\bar{\boldsymbol{x}}_t,\bar{\boldsymbol{x}}_0)}\left[\log\frac{p_\phi(\bar{\boldsymbol{x}}_{t-1}|\bar{\boldsymbol{x}}_t,\boldsymbol{c})}{p_{\boldsymbol{\theta}_{\text{ref}}}(\bar{\boldsymbol{x}}_{t-1}|\bar{\boldsymbol{x}}_t,\boldsymbol{c})}\right] + C \tag{19}$$

$$=\beta \sum_{t=1}^{T} \mathbb{E}_{\boldsymbol{c}\sim\mathcal{C},\bar{\boldsymbol{x}}_0\sim\mathcal{D}(\boldsymbol{c}),\bar{\boldsymbol{x}}_t\sim q(\bar{\boldsymbol{x}}_t|\bar{\boldsymbol{x}}_0),\bar{\boldsymbol{x}}_{t-1}\sim q(\bar{\boldsymbol{x}}_{t-1}|\bar{\boldsymbol{x}}_t,\bar{\boldsymbol{x}}_0)} \left[ \log \frac{p_\phi(\bar{\boldsymbol{x}}_{t-1}|\bar{\boldsymbol{x}}_t,\boldsymbol{c})}{q(\bar{\boldsymbol{x}}_{t-1}|\bar{\boldsymbol{x}}_t,\bar{\boldsymbol{x}}_0)} \right.$$

$$\left. + \log \frac{q(\bar{\boldsymbol{x}}_{t-1}|\bar{\boldsymbol{x}}_t,\bar{\boldsymbol{x}}_0)}{p_{\boldsymbol{\theta}_{\text{ref}}}(\bar{\boldsymbol{x}}_{t-1}|\bar{\boldsymbol{x}}_t,\boldsymbol{c})} \right] + C \qquad (20)$$

$$=\beta \sum_{t=1}^{T} \mathbb{E}_{\boldsymbol{c}\sim\mathcal{C},\bar{\boldsymbol{x}}_0\sim\mathcal{D}(\boldsymbol{c}),\bar{\boldsymbol{x}}_t\sim q(\bar{\boldsymbol{x}}_t|\bar{\boldsymbol{x}}_0)} \left[ -\mathbb{D}_{\text{KL}}\Big[q(\bar{\boldsymbol{x}}_{t-1}|\bar{\boldsymbol{x}}_t,\bar{\boldsymbol{x}}_0)\Big\|p_\phi(\bar{\boldsymbol{x}}_{t-1}|\bar{\boldsymbol{x}}_t,\boldsymbol{c})\Big] \right.$$

$$\left. + \mathbb{D}_{\text{KL}}\Big[q(\bar{\boldsymbol{x}}_{t-1}|\bar{\boldsymbol{x}}_t,\bar{\boldsymbol{x}}_0)\Big\|p_{\boldsymbol{\theta}_{\text{ref}}}(\bar{\boldsymbol{x}}_{t-1}|\bar{\boldsymbol{x}}_t,\boldsymbol{c})\Big] \right] + C \qquad (21)$$

$$=\beta \sum_{t=1}^{T} \frac{\beta_t^2}{2\sigma_t^2\alpha_t(1-\bar{\alpha}_t)} \mathbb{E}_{\boldsymbol{c}\sim\mathcal{C},\bar{\boldsymbol{x}}_0\sim\mathcal{D}(\boldsymbol{c}),\bar{\epsilon}\sim\mathcal{N}(\mathbf{0},\boldsymbol{I})} \left[ -\big\|\bar{\epsilon}-\epsilon_\phi(\bar{\boldsymbol{x}}_t,\boldsymbol{c},t)\big\|^2 \right.$$

$$\left. + \big\|\bar{\epsilon}-\epsilon_{\theta_{\text{ref}}}(\bar{\boldsymbol{x}}_t,\boldsymbol{c},t)\big\|^2 \right] + C \qquad (22)$$

$$=\beta \sum_{t=1}^{T} \lambda_t \mathbb{E}_{\boldsymbol{c}\sim\mathcal{C},\bar{\boldsymbol{x}}_0\sim\mathcal{D}(\boldsymbol{c}),\bar{\epsilon}\sim\mathcal{N}(\mathbf{0},\boldsymbol{I})} \left[ -\big\|\bar{\epsilon}-\epsilon_\phi(\bar{\boldsymbol{x}}_t,\boldsymbol{c},t)\big\|^2 + \big\|\bar{\epsilon}-\epsilon_{\theta_{\text{ref}}}(\bar{\boldsymbol{x}}_t,\boldsymbol{c},t)\big\|^2 \right] + C. \qquad (23)$$

All diffusion hyperparameter notations, i.e., $\sigma_t$, $\alpha_t$, $\bar{\alpha}_t$, and $\beta_t$, follow the definitions from DDPM [13]. Here, $\beta$ represents the KL regularization weight as defined in Eq. (1) and $C$ is a constant independent of $\phi$. We then derive the RHS:

$$\mathbb{E}_{\boldsymbol{c}\sim\mathcal{C},\boldsymbol{x}_{0:T}\sim p_{\boldsymbol{\theta}}(\boldsymbol{x}_{0:T}|\boldsymbol{c})} \left[ \beta \log \frac{p_\phi(\boldsymbol{x}_{0:T}|\boldsymbol{c})}{p_{\boldsymbol{\theta}_{\text{ref}}}(\boldsymbol{x}_{0:T}|\boldsymbol{c})} \right] \qquad (24)$$

$$=\mathbb{E}_{\boldsymbol{c}\sim\mathcal{C},\boldsymbol{x}_{0:T}\sim p_{\boldsymbol{\theta}}(\boldsymbol{x}_{0:T}|\boldsymbol{c})} \left[ \sum_{t=1}^{T} \beta \log \frac{p_\phi(\boldsymbol{x}_{t-1}|\boldsymbol{x}_t,\boldsymbol{c})}{p_{\boldsymbol{\theta}_{\text{ref}}}(\boldsymbol{x}_{t-1}|\boldsymbol{x}_t,\boldsymbol{c})} + \beta \log \frac{p_\phi(\boldsymbol{x}_T|\boldsymbol{c})}{p_{\boldsymbol{\theta}_{\text{ref}}}(\boldsymbol{x}_T|\boldsymbol{c})} \right] + C \qquad (25)$$

$$=\beta \sum_{t=1}^{T} \mathbb{E}_{\boldsymbol{c}\sim\mathcal{C},\boldsymbol{x}_t\sim p_{\boldsymbol{\theta}}(\boldsymbol{x}_t|\boldsymbol{c}),\boldsymbol{x}_{t-1}\sim p_{\boldsymbol{\theta}}(\boldsymbol{x}_{t-1}|\boldsymbol{x}_t,\boldsymbol{c})} \left[ \log \frac{p_\phi(\boldsymbol{x}_{t-1}|\boldsymbol{x}_t,\boldsymbol{c})}{p_{\boldsymbol{\theta}_{\text{ref}}}(\boldsymbol{x}_{t-1}|\boldsymbol{x}_t,\boldsymbol{c})} \right] + C \qquad (26)$$

$$=\beta \sum_{t=1}^{T} \mathbb{E}_{\boldsymbol{c}\sim\mathcal{C},\boldsymbol{x}_t\sim p_{\boldsymbol{\theta}}(\boldsymbol{x}_t|\boldsymbol{c}),\boldsymbol{x}_{t-1}\sim p_{\boldsymbol{\theta}}(\boldsymbol{x}_{t-1}|\boldsymbol{x}_t,\boldsymbol{c})} \left[ \log \frac{p_\phi(\boldsymbol{x}_{t-1}|\boldsymbol{x}_t,\boldsymbol{c})}{p_{\boldsymbol{\theta}}(\boldsymbol{x}_{t-1}|\boldsymbol{x}_t,\boldsymbol{c})} + \log \frac{p_{\boldsymbol{\theta}}(\boldsymbol{x}_{t-1}|\boldsymbol{x}_t,\boldsymbol{c})}{p_{\boldsymbol{\theta}_{\text{ref}}}(\boldsymbol{x}_{t-1}|\boldsymbol{x}_t,\boldsymbol{c})} \right] + C$$
$$(27)$$

$$=\beta \sum_{t=1}^{T} \mathbb{E}_{\boldsymbol{c}\sim\mathcal{C},\boldsymbol{x}_t\sim p_{\boldsymbol{\theta}}(\boldsymbol{x}_t|\boldsymbol{c})} \left[ -\mathbb{D}_{\text{KL}}\Big[p_{\boldsymbol{\theta}}(\boldsymbol{x}_{t-1}|\boldsymbol{x}_t,\boldsymbol{c})\Big\|p_\phi(\boldsymbol{x}_{t-1}|\boldsymbol{x}_t,\boldsymbol{c})\Big] + \right.$$

$$\left. \mathbb{D}_{\text{KL}}\Big[p_{\boldsymbol{\theta}}(\boldsymbol{x}_{t-1}|\boldsymbol{x}_t,\boldsymbol{c})\Big\|p_{\boldsymbol{\theta}_{\text{ref}}}(\boldsymbol{x}_{t-1}|\boldsymbol{x}_t,\boldsymbol{c})\Big] \right] + C \qquad (28)$$

$$=\beta \sum_{t=1}^{T} \frac{\beta_t^2}{2\sigma_t^2\alpha_t(1-\bar{\alpha}_t)} \mathbb{E}_{\boldsymbol{c}\sim\mathcal{C},\boldsymbol{x}_t\sim p_{\boldsymbol{\theta}}(\boldsymbol{x}_t|\boldsymbol{c})} \left[ -\big\|\boldsymbol{\epsilon_\theta}(\boldsymbol{x}_t,\boldsymbol{c},t)-\boldsymbol{\epsilon_\phi}(\boldsymbol{x}_t,\boldsymbol{c},t)\big\|^2 \right.$$

$$\left. + \big\|\boldsymbol{\epsilon_\theta}(\boldsymbol{x}_t,\boldsymbol{c},t)-\boldsymbol{\epsilon_{\theta_{\text{ref}}}}(\boldsymbol{x}_t,\boldsymbol{c},t)\big\|^2 \right] + C \qquad (29)$$

$$=\beta \sum_{t=1}^{T} \lambda_t \mathbb{E}_{\boldsymbol{c}\sim\mathcal{C},\boldsymbol{x}_t\sim p_{\boldsymbol{\theta}}(\boldsymbol{x}_t|\boldsymbol{c})} \left[ -\big\|\boldsymbol{\epsilon_\theta}(\boldsymbol{x}_t,\boldsymbol{c},t)-\boldsymbol{\epsilon_\phi}(\boldsymbol{x}_t,\boldsymbol{c},t)\big\|^2 \right.$$

$$\left. \big\|\boldsymbol{\epsilon_\theta}(\boldsymbol{x}_t,\boldsymbol{c},t)-\boldsymbol{\epsilon_{\theta_{\text{ref}}}}(\boldsymbol{x}_t,\boldsymbol{c},t)\big\|^2 \right] + C. \qquad (30)$$

Substituting Eqs. (23) and (30) into Eq. (16), we obtain:

$$
\sum_{t=1}^{T} \lambda_t \mathbb{E}_{\boldsymbol{c}\sim\mathcal{C},\bar{\boldsymbol{x}}_0\sim\mathcal{D}(\boldsymbol{c}),\bar{\epsilon}\sim\mathcal{N}(\boldsymbol{0},\boldsymbol{I})} \Big[ -\big\|\bar{\epsilon}-\epsilon_\phi(\bar{\boldsymbol{x}}_t,\boldsymbol{c},t)\big\|^2 + \big\|\bar{\epsilon}-\epsilon_{\theta_{\text{ref}}}(\bar{\boldsymbol{x}}_t,\boldsymbol{c},t)\big\|^2 \Big]
$$

$$
-\sum_{t=1}^{T} \lambda_t \mathbb{E}_{\boldsymbol{c}\sim\mathcal{C},\boldsymbol{x}_t\sim p_{\boldsymbol{\theta}}(\boldsymbol{x}_t|\boldsymbol{c})} \Big[ -\big\|\boldsymbol{\epsilon}_{\boldsymbol{\theta}}(\boldsymbol{x}_t,\boldsymbol{c},t)-\boldsymbol{\epsilon}_\phi(\boldsymbol{x}_t,\boldsymbol{c},t)\big\|^2 + \big\|\boldsymbol{\epsilon}_{\boldsymbol{\theta}}(\boldsymbol{x}_t,\boldsymbol{c},t)-\boldsymbol{\epsilon}_{\boldsymbol{\theta}_{\text{ref}}}(\boldsymbol{x}_t,\boldsymbol{c},t)\big\|^2 \Big]
$$

$$(31)$$

Following the DDPM settings, we set $\lambda_t = 1$ to obtain our final result.

## E  Ethics

The Pick-a-Pic v2 dataset has been identified as containing NSFW prompts, as it is collected from publicly available user inputs on the internet. To minimize exposure to violent, adult, or otherwise inappropriate content, we chose HPDv2 as the prompt source for user studies. Participants are also informed of this facts before they start the study.

For a fair comparison with previous methods, we continue to use Pick-a-Pic v2 as part of the training data. Given the strong performance of Diffusion-DRO, there is a potential risk that the model could generate NSFW content. However, Diffusion-DRO does not explicitly learn to produce NSFW images; its outputs are inherently dependent on the training dataset.

To mitigate this risk in future applications, NSFW content can be filtered at the data level by curating human preference datasets that exclude inappropriate content, thereby preventing Diffusion-DRO from learning to generate such images. Before publicly releasing our model, we will ensure the implementation of an additional safety filter to prevent misuse.

## F  Limitations

Despite the significant improvements Diffusion-DRO brings to aligning diffusion models with human preferences, it remains constrained by the data-dependent nature of diffusion models. Specifically, the approach relies on expert demonstrations extracted from data, which may introduce distributional biases—for example, simpler prompts tend to yield better outputs and are thus more likely to be selected as expert data. Diffusion-DRO does not explicitly account for such biases and disregards non-expert demonstrations, which may result in a model that performs well only in limited domains.

## G  Future Work

While Diffusion-DRO introduces the concept of expert demonstrations from an inverse reinforcement learning perspective, future work could extend this framework beyond the current max-margin formulation by incorporating non-expert data to enhance performance in underrepresented or sparse regions of the data distribution. Furthermore, our current approach treats preferences as binary rankings (i.e., preferred vs. not preferred), which results in the loss of list-wise ranking information [19]. We believe that integrating such richer preference structures into the inverse reinforcement learning framework could further refine the granularity and stability of the optimization process.

# H   Additional Samples

In this section, we provide additional sample comparisons with baseline methods, including Diffusion-DPO w/ SFT, Diffusion KTO w/ SFT, Diffusion-SPO, and SD v1-5.

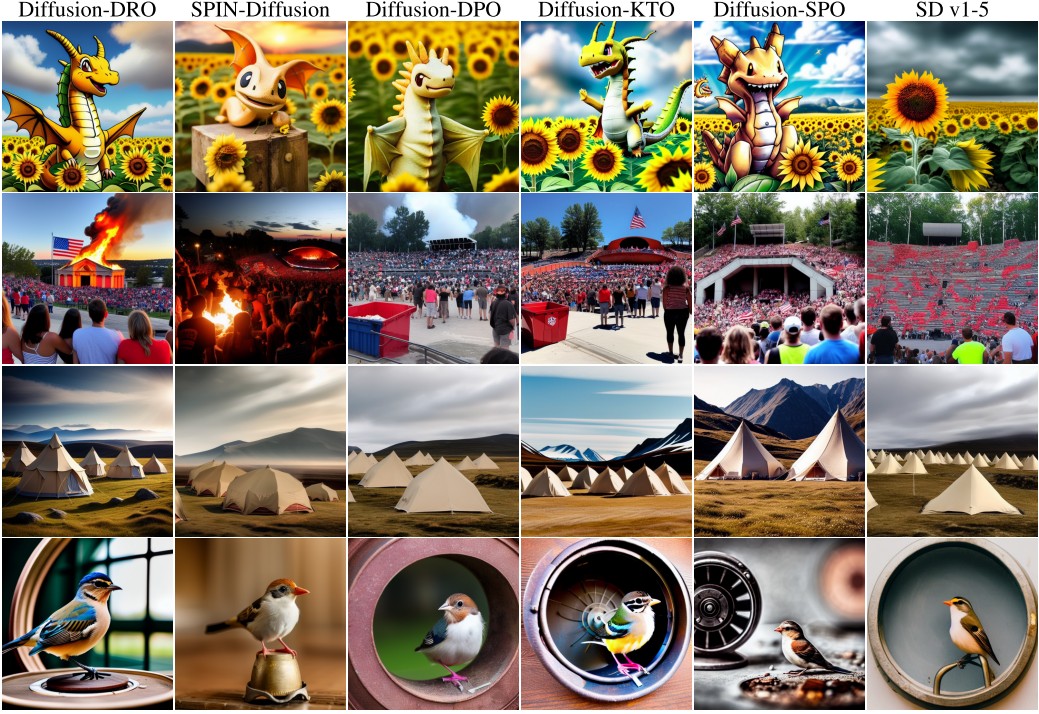

Figure 4: The prompts used for image generation are sourced from the HPDv2 Benchmark, categorized as Anime, Concept Art, Painting, and Photo from top to bottom, respectively. The specific prompts, in order, are: "*A portrait of a smiling Dragonite in a sunflower field with a cloudy sky backdrop,*" "*Amphitheater filled with crowd looking at a dumpster on fire in patriotic colors,*" "*Beige canvas tents set up in an arctic landscape with no vegetation, surrounded by rolling hills - reminiscent of a romanticist painting,*" and "*A small bird sitting in a metal wheel.*"

| Diffusion-DRO | SPIN-Diffusion | Diffusion-DPO | Diffusion-KTO | Diffusion-SPO | SD v1-5 |

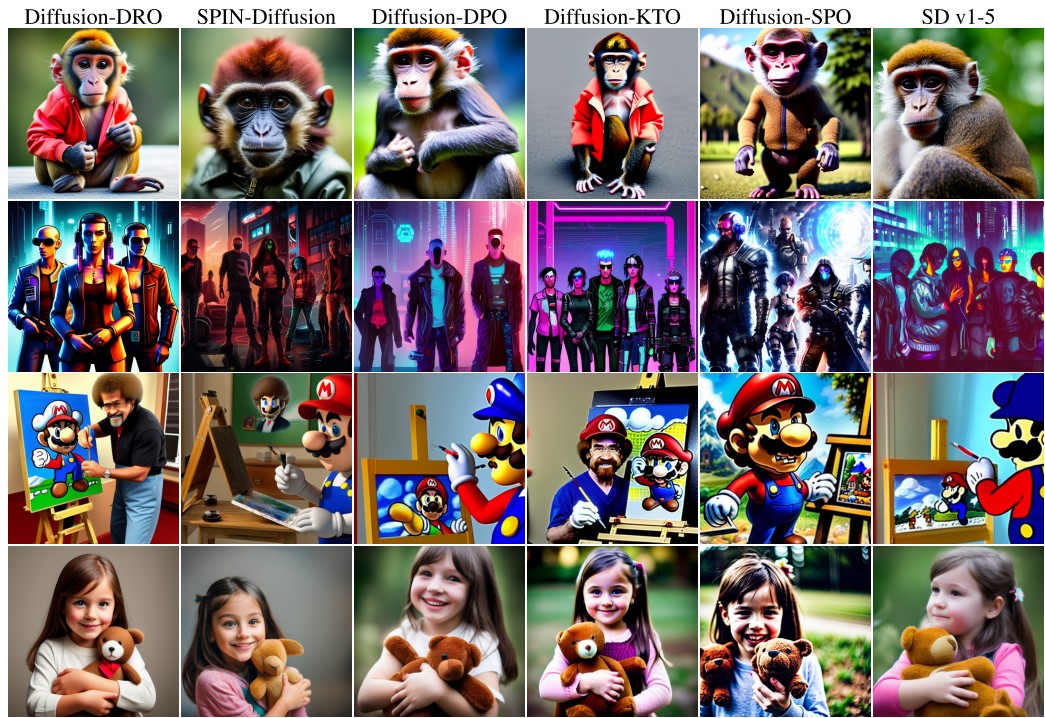

Figure 5: The prompts used for image generation are sourced from the HPDv2 Benchmark, categorized as Anime, Concept Art, Painting, and Photo from top to bottom, respectively. The specific prompts, in order, are: "*A monkey wearing a jacket,*" "*Portrait of a cyberpunk gang,*" "*Bob Ross painting Mario on an easel in his office,*" and "*A little girl holding a brown stuffed animal.*"

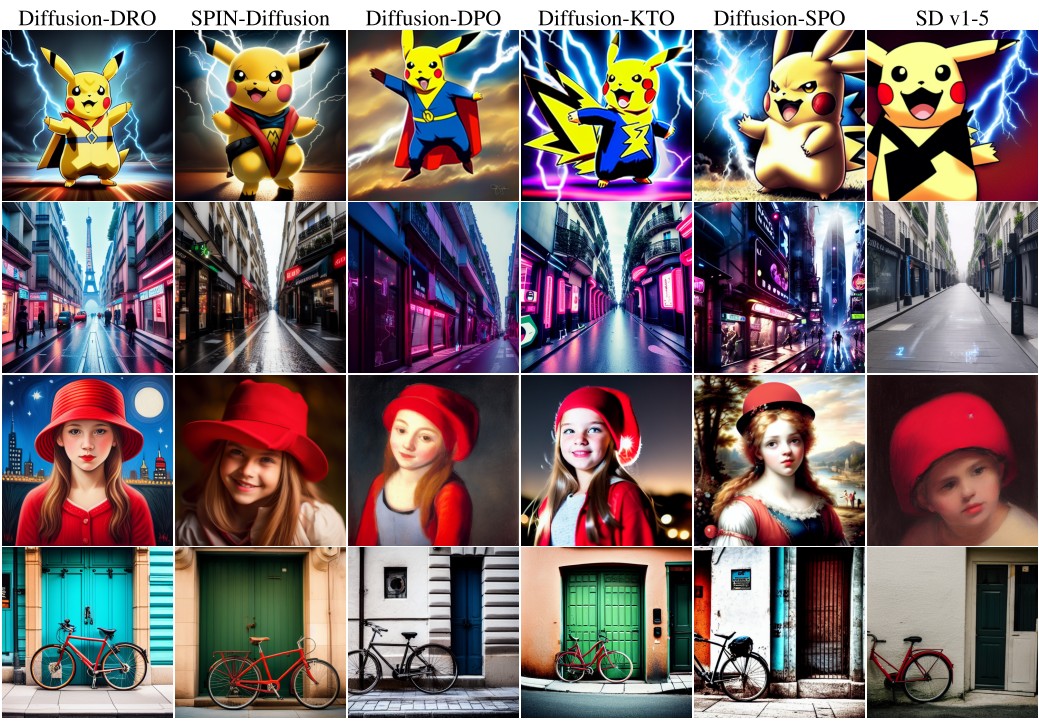

Figure 6: The prompts used for image generation are sourced from the HPDv2 Benchmark, categorized as Anime, Concept Art, Painting, and Photo from top to bottom, respectively. The specific prompts, in order, are: *"A new artwork depicting Pikachu as a superhero fighting villains with dramatic lightning,"* *"A futuristic cyberpunk Paris street,"* *"A young girl with a red hat at night,"* and *"A bike parked in front of a doorway."*

