# OpenReview forum: "Ranking-based Preference Optimization for Diffusion Models from Implicit User Feedback"
_NeurIPS.cc/2025/Conference — NeurIPS 2025 poster_

### Official Review · Reviewer_Fj6s · 2025-06-25

**Clarity:** 3
**Significance:** 2
**Originality:** 3
**Rating:** 4
**Confidence:** 3

**Summary:**

The authors propose Diffusion-DRO, a novel algorithm inspired by Inverse Reinforcement Learning for aligning text-to-image (T2I) diffusion models with human preferences using expert knowledge. Diffusion-DRO employs a minimax optimization framework where a reward model maximizes the margin between expert and policy generations, while the policy network minimizes this margin. The authors validate their approach through experiments on Stable Diffusion v1.5, demonstrating superior performance compared to baseline alignment methods such as KTO and SPO using both automatic metrics and human evaluation.

**Questions:**

* As mentioned in **Strengths And Weaknesses** section, could the authors provide preliminary results on the generalizability of the DRO on larger size models, say SDXL if time permits.
* It appears to me that the threshold $m$ is an crucial factor in preventing overfitting, could the authors provide some ablation studies on the behavior of Diffusion-DRO w.r.t different choices of m?
* It seems a bit unnatural that the aesthetics focused scores are dropping as the training set size increases with both SFT and DRO, given that in table 1 SFT improves such scores for SD1.5. could the authors elaborate a bit more on this?

**Ethical Concerns:**

["NO or VERY MINOR ethics concerns only"]

**Final Justification:**

As mentioned in comments

**Limitations:**

yes

**Paper Formatting Concerns:**

I don't see formatting issue

**Quality:**

3

**Strengths And Weaknesses:**

* Strengths:
  * Novel theoretical perspective from Inverse RL. The Max Margin loss can be simplified to an intuitive form of SFT with winning data plus contrastive loss between policy and reward. The overall methodological contribution is solid.
  * Comprehensive experiments for both automatic and human preference evaluation support the authors' claims.

  * Clear writing with detailed evaluation protocols ensuring fair comparison with baselines.

* Weakness:
  * My major concern is the generalizability of Diffusion-DRO. The Diffusion-DRO loss handles winning images from expert data, which could be very effective on medium-sized T2I models like SD1.5. However, [1] suggests that SFT with Pick-a-Pic v2 causes performance degradation on SDXL. This indicates we might need higher-quality expert data to scale up to slightly larger models. It would be interesting if the authors could provide evidence that Diffusion-DRO does not hurt SDXL performance within a few optimization steps.
  * Another potential issue with Diffusion-DRO is tuning, the push-away term $||\epsilon_{\theta}(x_t, c,t) - \epsilon_{\phi}(x_t, c,t) ||_2^2$ could "push too far" and cause overfitting/collapse when the SFT loss component saturates, tackling this issue with different T2I models could require significant tuning efforts on hyper-parameters


[1]  Diffusion model alignment using direct preference optimization. Wallace et,al, 2024

---

> ### Author Rebuttal · Authors · 2025-07-31
>
> We sincerely thank the reviewer for the valuable feedback. Below, we address each of the questions in detail. As the concerns raised in the weaknesses section are directly reflected in these questions, we focus our response on answering the questions provided.
>
> ---
>
> - ### Q1
>     As mentioned in Strengths And Weaknesses section, could the authors provide preliminary results on the generalizability of the DRO on larger size models, say SDXL if time permits.
>     ###
>     ### A1
>     In response to a suggestion from reviewer `NjRJ`, we extended our experiments to a larger model and dataset, specifically SDXL and MJHQ-30k [A]. We also adopted the GenEval benchmark [B], as proposed by NjRJ, to more comprehensively assess model performance.
>
>     We ranked MJHQ-30k samples using PickScore and selected the top 500 image-text pairs as expert demonstrations. Both `SDXL + SFT` and `Diffusion-DRO` were fine-tuned on this dataset. For reference, we also report results from the original `SDXL` model and `Diffusion-DPO`, the latter of which was fine-tuned on Pick-a-Pic v2. Following the GenEval protocol, each prompt was used to generate four images to ensure reliable evaluation.
>
>     | Method        |   Colors  | Color Attr. | Single Object | Two Object |  Counting |  Position |  Overall  |
>     | ------------- | :-------: | :---------: | :-----------: | :--------: | :-------: | :-------: | :-------: |
>     | SDXL          |   87.23   |    23.00    |     99.69     |    78.03   |   43.75   |   11.75   |   57.24   |
>     | Diffusion-DPO |   86.17   |    25.00    |     98.75     |    85.25   |   48.25   | **13.60** |   59.50   |
>     | SDXL + SFT    |   88.94   |  **27.00**  |     99.75     |    87.68   |   46.50   |   13.20   |   60.51   |
>     | Diffusion-DRO | **90.43** |  **27.00**  |   **100.00**  |  **87.88** | **50.00** |   12.00   | **61.22** |
>
>     This demonstrates the method’s strong generalization ability to both a larger backbone (SDXL) and a higher-quality dataset (MJHQ-30k). Notably, even when fine-tuning on expert demonstrations in the form of image-text pairs alone, without explicit preference labels, Diffusion-DRO is able to outperform SFT. For example, it achieves a 1.49 percent gain in the "Colors" category and a 3.5 percent improvement in "Counting."
>
>     [A] "Playground v2.5: Three Insights towards Enhancing Aesthetic Quality in Text-to-Image Generation", arXiv.
>
>     [B] "GenEval: An Object-Focused Framework for Evaluating Text-to-Image Alignment." NeurIPS 2023.
>
>     ---
>
> - ### Q2
>     It appears to me that the threshold is an crucial factor in preventing overfitting, could the authors provide some ablation studies on the behavior of Diffusion-DRO w.r.t different choices of m?
>     ###
>     ### A3
>     We fully agree that the threshold $𝑚$ can play a critical role in preventing the model from "pushing too far." Below, we provide an ablation study evaluating Diffusion-DRO with different values of $m$ on the Pick-a-Pic v2 test set, keeping all other settings consistent with those in Section 4.
>
>     | Method                      |PickScore|Aesthetic |CLIP       |IR        |
>     |:----------------------------|:-------:|:--------:|:---------:|:--------:|
>     | Diffusion-DPO               |21.31    |6.35      |0.3334     |0.7912    |
>     | Diffusion-KTO               |21.25    |6.34      |0.3296     |0.7636    |
>     | SPIN-Diffusion              |21.40    |6.26      |0.3305     |0.5619    |
>     | Diffusion-DRO ($m=10^{0}$)  |21.37    |6.38      |0.3290     |0.7279    |
>     | Diffusion-DRO ($m=10^{-1}$) |21.39    |6.40      |0.3267     |0.7076    |
>     | Diffusion-DRO ($m=10^{-2}$) |21.49    |**6.47**  |0.3268     |0.7399    |
>     | Diffusion-DRO ($m=10^{-3}$ used in paper) |**21.54**|**28.48**|6.42      |0.3390     |**0.8452**|
>     | Diffusion-DRO ($m=10^{-4}$) |21.29    |6.25      |0.3431     |0.7104    |
>     | Diffusion-DRO ($m=0$)       |21.13    |6.18      |**0.3426** |0.5955    |
>
>     From the results, we observe that when $m$ becomes extremely small, such as $10^{-4}$ or $0$, performance drops due to the "push too far" effect. The best performance is achieved at $m = 10^{-3}$, and while increasing $m$ further slightly reduces the scores, Diffusion-DRO still consistently outperforms both Diffusion-DPO and Diffusion-KTO across most metrics.
>
>     We hope this ablation provides a clearer understanding of how the threshold affects model performance.
>
>     ---
>
> - ### Q3
>     It seems a bit unnatural that the aesthetics focused scores are dropping as the training set size increases with both SFT and DRO, given that in table 1 SFT improves such scores for SD1.5. could the authors elaborate a bit more on this?
>     ###
>     ### A3
>     Thank you for raising this point. We believe you are referring to the trend shown in Figure 3, but please feel free to correct us if we misunderstood.
>
>     In this figure, the so-called training set is not a random subset of the full dataset. Instead, it is constructed by ranking the training data using the HPSv2 preference model and selecting the top K samples. As K increases, the average quality of the selected expert demonstrations gradually decreases. This leads to a drop in performance for both SFT and DRO, especially on metrics related to aesthetics.
>
>     To avoid confusion, we will revise the x-axis label in the subfigures of Figure 3 from "Train Size" to "Top K" to better reflect that it refers to the size of the expert demonstration subset, ranked by preference quality.

---

> > ### Comment · Reviewer_Fj6s · 2025-08-04
> >
> > I appreciate the author's efforts in the rebuttal. In A3 row 7 of the table, I guess the Aesthetic and CLIP score are flipped. However, I remain unconvinced about the generalizability of DRO to more recent text-to-image diffusion architectures such as MMDiT and SANA, although I understand that implementing these extensions could be time-consuming.  Therefore, I will keep my current rating.

---

> > > ### Author Response · Authors · 2025-08-05
> > >
> > > Thank you for your thoughtful feedback and for pointing out the inconsistency in A3 row 7. We sincerely appreciate your careful reading. Upon revisiting the table, we found that a misalignment occurred due to an erroneous cell entry, which caused the values starting from the Aesthetic score to shift one column to the right. We have corrected the table accordingly. Only the row for **Diffusion-DRO (\$m=10^{-3}\$ used in paper)** has been modified:
> > >
> > > | Method                                      | PickScore | Aesthetic |    CLIP    |     IR     |
> > > | :------------------------------------------ | :-------: | :-------: | :--------: | :--------: |
> > > | Diffusion-DPO                               |   21.31   |    6.35   |   0.3334   |   0.7912   |
> > > | Diffusion-KTO                               |   21.25   |    6.34   |   0.3296   |   0.7636   |
> > > | SPIN-Diffusion                              |   21.40   |    6.26   |   0.3305   |   0.5619   |
> > > | Diffusion-DRO (\$m=10^{0}\$)                |   21.37   |    6.38   |   0.3290   |   0.7279   |
> > > | Diffusion-DRO (\$m=10^{-1}\$)               |   21.39   |    6.40   |   0.3267   |   0.7076   |
> > > | Diffusion-DRO (\$m=10^{-2}\$)               |   21.49   |  **6.47** |   0.3268   |   0.7399   |
> > > | Diffusion-DRO (\$m=10^{-3}\$ used in paper) | **21.54** |    6.42   |   0.3390   | **0.8452** |
> > > | Diffusion-DRO (\$m=10^{-4}\$)               |   21.29   |    6.25   |   0.3431   |   0.7104   |
> > > | Diffusion-DRO (\$m=0\$)                     |   21.13   |    6.18   | **0.3426** |   0.5955   |

---

### Official Review · Reviewer_Y9sn · 2025-06-25

**Clarity:** 2
**Significance:** 2
**Originality:** 2
**Rating:** 4
**Confidence:** 2

**Summary:**

This paper is quite formula-heavy, so I’m trying to clarify its main points. The authors aim to improve the performance of diffusion models using reinforcement learning (RL). The abstract notes that previous methods improved training stability by avoiding the REINFORCE algorithm but still face challenges such as accurately estimating image probabilities due to the nonlinear nature of the sigmoid function and the limited diversity of offline datasets.

The main challenges are thus “accurately estimating image probabilities” (which I interpret as accurately estimating the reward for images) and the limited diversity of offline datasets. The abstract then proposes a method called Diffusion-DRO. By casting preference learning as a ranking problem, this approach overcomes the nonlinear estimation issues. It also integrates offline expert demonstrations with online policy-generated negative samples to address the limited diversity of offline datasets.

Now, looking at how Diffusion-DRO achieves these two goals: equations (1) to (6) explain the basics of RL and reward functions, but equations (7) to (12) are quite obscure. My understanding is that the authors want there to exist an oracle image from the distribution D(c), and the model-generated images should be as close as possible in quality to those from D(c). However, if this were the whole story, wouldn’t simple supervised fine-tuning (SFT) suffice?

Then, equation (13) mentions there is an extra term compared to SFT. The authors say that the second term serves a complementary purpose, where the reward model generates negative samples online to guide the optimization in the correct direction. I don’t quite understand this part.

According to my shallow understanding, the SFT term encourages the model’s current noise prediction to match the ground truth noise, while the second term encourages the current model’s noise prediction to differ from that of its previous version, which helps ensure diversity. I think I understand how this addresses the limited diversity of offline datasets. However, I am still unsure whether this fully solves the problem of accurately estimating image probabilities. This seems to relate to the ranking formulation, but I am struggling to grasp how ranking is used here.

Jumping ahead to the experiments, the authors demonstrate the effectiveness of Diffusion-DRO by showing win rates through pairwise comparisons. The results indicate that Diffusion-DRO outperforms other models on most metrics.

**Questions:**

Sorry, I actually didn’t quite understand the paper. I hope the authors can provide a more straightforward and easy-to-understand explanation.

**Ethical Concerns:**

["NO or VERY MINOR ethics concerns only"]

**Final Justification:**

Thank you for your reply; it has resolved all my doubts. I will raise my scores accordingly.

**Limitations:**

yes

**Quality:**

3

**Strengths And Weaknesses:**

[+] Good performance.

[-] Please refer to the summary, there are still many things I don't understand.

[-] Although the article says that the reward model can be removed, Algorithm 1 still needs to initialize a reward model.

[-] The ICML 2024 paper Self-Play Fine-Tuning Converts Weak Language Models to Strong Language Models treats the SFT dataset as oracle data, which is equivalent to the expert demonstrations in this context. In their approach, any content generated by the model is regarded as a negative example, while the SFT examples are treated as positive examples—similar to the second term in equation (13).

---

> ### Author Rebuttal · Authors · 2025-07-31
>
> We sincerely thank the reviewer for the careful and thoughtful review. Below, we respond to the concerns raised in both the weaknesses and questions sections.
>
> ---
>
> - ### Q1
>     Now, looking at how Diffusion-DRO achieves these two goals: equations (1) to (6) explain the basics of RL and reward functions, but equations (7) to (12) are quite obscure. My understanding is that the authors want there to exist an oracle image from the distribution $D(c)$, and the model-generated images should be as close as possible in quality to those from $D(c)$. However, if this were the whole story, wouldn’t simple supervised fine-tuning (SFT) suffice?
>     ###
>     ### A1
>
>     From a reinforcement learning perspective, inverse reinforcement learning (IRL) offers a principled alternative to supervised fine-tuning (SFT), which is also known as behavior cloning in RL terminology. We summarize the key differences in the following table:
>
>     | Aspect        | SFT               | Inverse RL |
>     | ------------- | ------------------------------------------ | ------------------------------------ |
>     | Objective     | Imitate expert policy                      | Recover reward function              |
>     | Required Data | Expert text-image pairs                    | Expert text-image pairs              |
>     | Advantages    | Simple to implement; no interaction required | Interpretable objectives; better generalization |
>     | Disadvantages | Susceptible to compounding errors; limited extrapolation | Harder to optimize; sample inefficient  |
>
>     In summary, we choose inverse RL over direct supervised fine-tuning in order to improve generalization. Rather than directly imitating expert demonstrations, our method learns a reward model that implicitly evaluates generation quality. This enables stronger learning signals and allows the policy to generalize beyond the specific modes captured in the demonstration data.
>
>     ---
>
> - ### Q2
>     According to my shallow understanding, the SFT term encourages the model’s current noise prediction to match the ground truth noise, while the second term encourages the current model’s noise prediction to differ from that of its previous version, which helps ensure diversity. I think I understand how this addresses the limited diversity of offline datasets. However, I am still unsure whether this fully solves the problem of accurately estimating image probabilities. This seems to relate to the ranking formulation, but I am struggling to grasp how ranking is used here.
>     ###
>     ### A2
>     The first term (SFT-style loss) indeed encourages accurate noise prediction, while the second term pushes the reward model to assign lower scores to policy-generated samples, helping to diversify and improve generation quality.
>
>     The core of our contribution lies in the **ranking-based formulation** introduced in Eq. 7, which specifies that expert demonstrations should receive higher rewards than policy samples. This ranking constraint is enforced via a **max-margin objective**, allowing the reward model to learn preference-consistent scores without requiring exact image likelihoods. Unlike Diffusion-DPO, which uses a non-linear transformation $\log \sigma(\cdot)$ to convert rewards into classification logits (Eq. 14), our method **maintains a linear relationship** between expert and policy scores. This avoids the use of Jensen’s inequality, which introduces a relaxation gap in the optimization. As a result, our approach can **more precisely optimize the reward margin**, leading to stronger alignment with expert demonstrations.
>
>     While we do not attempt to estimate exact image probabilities, which is known to be difficult in diffusion models, our ranking loss offers a more **faithful training signal** for learning sample quality preferences. This directly addresses the challenge of aligning generation quality with human-like judgments without relying on brittle probabilistic modeling.
>
>     We hope this clarifies how ranking is directly utilized in our formulation to improve the fidelity of reward modeling.
>
>     ---
>
> - ### Q3
>     Although the article says that the reward model can be removed, Algorithm 1 still needs to initialize a reward model.
>     ###
>     ### A3
>     In conventional reinforcement learning from human feedback (RLHF), a reward model is typically trained on preference data first, and then used to fine-tune the generative model. This results in a two-stage training pipeline.
>
>     In contrast, our method does not require pretraining a separate reward model. As shown in Eq. 10, our reward model and policy model are essentially equivalent. Therefore, we only need to train the policy model once, and the reward signal is implicitly defined through this joint formulation. This allows us to bypass the need for a standalone reward model and simplifies the training process into a single stage.
>
>     ---
>
> - ### Q4
>     The ICML 2024 paper Self-Play Fine-Tuning Converts Weak Language Models to Strong Language Models treats the SFT dataset as oracle data, which is equivalent to the expert demonstrations in this context. In their approach, any content generated by the model is regarded as a negative example, while the SFT examples are treated as positive examples—similar to the second term in equation (13).
>     ###
>     ### A4
>     While both SPIN [A] and SPIN-Diffusion [B] make use of oracle data, the notion of expert demonstrations in our work differs in a fundamental way. Specifically, our expert demonstrations are required to satisfy the max-margin condition in Eq. 7, meaning that their quality must be optimal with respect to the reward model.
>
>     In contrast, SPIN and SPIN-Diffusion train a discriminator to distinguish whether a sample originates from the dataset or from the model. This is particularly evident in SPIN-Diffusion, where the discriminator must be updated frequently to maintain its effectiveness. As a result, their training procedure follows a *multi-stage* process.
>
>     Although our final formulations may appear similar, the role and interpretation of expert demonstrations in our method are conceptually distinct from the discriminators used in SPIN-based approaches.
>
>     [A] Self-Play Fine-Tuning Converts Weak Language Models to Strong Language Models, ICML 2024.
>
>     [B] Self-Play Fine-Tuning of Diffusion Models for Text-to-Image Generation, NeurIPS 2024.

---

> > ### Comment · Reviewer_Y9sn · 2025-08-04
> >
> > Thank you for the author's reply.
> >
> > I understand that Eq. 7 specifies that expert demonstrations should receive higher rewards than policy samples. Since expert demonstrations are considered the best, all policy-generated samples are treated as inferior. This reminds me of DPO, where the chosen samples are expert demonstrations and the rejected ones are policy samples. So, why does this paper not directly use DPO?

---

> > > ### Author Response · Authors · 2025-08-05
> > >
> > > It is indeed possible to apply DPO in this setting by iteratively updating the policy to ensure the quality of policy-generated samples. However, doing so would constrain the loss function to a specific binary classification form, namely the $\log \sigma(\cdot)$ objective. In contrast, our approach is derived from the perspective of max-margin inverse reinforcement learning, leading to Eq. 12, which offers a more general formulation. If one applies the $\log \sigma(\cdot)$ transformation to maximize the margin in Eq. 12, it effectively recovers the DPO objective—this is precisely what you referred to as "directly using DPO."
> > >
> > > Starting from this generalized form allows us to explore alternative loss functions that better optimize the margin structure. In particular, our thresholded ranking loss (Eq. 15) emerges from this perspective and has shown improved effectiveness in practice. We believe this broader view not only maintains compatibility with DPO but also offers greater flexibility and theoretical grounding.

---

> > > > ### Author Response · Authors · 2025-08-08
> > > >
> > > > Thank you for engaging with our explanations and for the discussion. We are glad if our clarification on the max-margin IRL formulation and its relationship to DPO was helpful. We would like to emphasize that starting from the generalized max-margin perspective (Eq. 12) is what enables our thresholded ranking loss (Eq. 15), which showed consistent improvements over DPO across all tested settings. This flexibility is not available if directly using DPO’s fixed objective. We hope this helps reinforce the core contribution and practical advantage of Diffusion-DRO. We truly appreciate your time and feedback.

---

### Official Review · Reviewer_82HS · 2025-07-02

**Clarity:** 3
**Significance:** 3
**Originality:** 3
**Rating:** 4
**Confidence:** 3

**Summary:**

* This paper proposes a novel method for aligning text-to-image diffusion models with human preferences. Instead of fine-tuning a policy by preference data, this paper proposes to fine-tune a policy by using expert demonstration data. To achieve, this, the paper proposes a new algorithm, Diffusion Denoising Ranking Optimization (Diffusion-DRO), that treat the RLHF (alignment) problem as apprenticeship learning, that alternate between training a reward function and training a policy (conceptually similar to GAIL (Ho and Ermon 2016)), but for RLHF with diffusion policies setting).

* The key difference between the proposed work and Supervised Fine-Tuning (SFT) in previous work is that it not only fit the reward model to match the expert demonstration as done by SFT, but also train the reward and policy in an adversarial manner, i.e., optimize policy to max the learned reward & learn a reward that damage the learned policy.

* The key difference between the proposed work and DPO-based approaches in previous work is that this work is doing apprenticeship learning (IRL + policy optimization) while DPO-based approaches is learning from preferences + policy optimization in 1 step.

* Results evaluate the proposed method in 2 different datasets (Pick-a-Pic v2 and HPDv2), against four baselines, including SPIN-Diffusion [40], Diffusion-SPO [18], Diffusion-DPO [34], and Diffusion-KTO [17]. Regarding the metrics, including PickScore, Aesthetic Score, ImageReward, and human preferences in a user study, the proposed method significantly outperformed baselines. Regarding the metric, CLIP Score, sometimes the proposed method is outperformed.

**Questions:**

* For line 102, the paper says that they make an assumption formulated in Eq.4. I am curious when this assumption holds. It seems that Eq.4 says that the reward of the image x0 given prompt c, depends on the entire diffusion process x_{1:T} that generates x0. This is a bit counterintuitive for me because I think whether an image x0 is good or bad only depends on itself, not its generation process. Could you help me ground this assumption a bit more?

* For line 127, the authors say that they parameterize the trajectory reward as similar to Eq.3. I wonder whether this is a choice made by the authors for math convenience, or perhaps there is some ground truth supporting this parameterization?


* For lines 226-231, I think the text tries to explain why the proposed method, Diffusion-DRO, is outperformed by baselines regarding the CLIP scores. But I don't quite understand the explanations from lines 226-231. Is the reason that Diffusion-DRO only uses the expert demonstrations, a small part of the preference dataset, resulting in a bad text encoder?

**Ethical Concerns:**

["NO or VERY MINOR ethics concerns only"]

**Final Justification:**

I think this paper contributes a new method for RLHF on diffusion models. After the rebuttal, I am still concerned that the algorithm needs high-quality demonstrations. In other words, it requires hand-designed metrics to find good demonstration data. Depending on the selection metrics, the amount of demo data will differ, making it hard to tune the K parameter. Moreover, from the table provided in the author's response, it seems that the improvement of Diffusion-DRO over Diffusion-DPO is quite small. It is unclear how significant that is in reality. So I plan to keep the score now.

**Limitations:**

Yes

**Quality:**

3

**Strengths And Weaknesses:**

* This paper solves a well-motivated problem, is well-written, proposes a novel method, and demonstrates with significantly good empirical results.

* One key concern of the proposed method is that it seems to lose the benefit of DPO methods that are designed to avoid first fitting a reward function and then finding a policy to optimize the reward. Instead, the proposed method in Alg.1 has to first do IRL (Alg.1 line 14) and then do policy optimization (Alg.1 line 16). It is very helpful that Sec.3.4 discusses the frequency of the two alternative steps for IRL and policy optimization. But I wonder whether the alternation between reward and policy learning results in increased computation overhead? Maybe some empirical evaluation of computation could be helpful to address this.
    * Also, it is not quite clear to me how the pϕ in Alg.1 line 16 is computed. It would be great if the authors could clarify this.

* Another concern is to find demonstration data. Preference data are largely available and easy for humans to provide. But expert demonstrations are hard to obtain. This paper proposes to use the top K images as the expert demonstrations, from a given preference dataset ranked by the preference labels. This makes sense, but introduces a hyperparameter K to tune.

---

> ### Author Rebuttal · Authors · 2025-07-31
>
> We sincerely appreciate your thoughtful review. Below, we address each of your questions in detail. Lastly, we provide clarifications for several concerns mentioned in the weaknesses section that were not explicitly formulated as questions.
>
> ## Reply to Questions
>
> ---
>
> - ### Q1
>     For line 102, the paper says that they make an assumption formulated in Eq. 4. I am curious when this assumption holds. It seems that Eq. 4 says that the reward of the image $x\_0$ given prompt $c$, depends on the entire diffusion process $x\_{1:T}$ that generates $x\_0$. This is a bit counterintuitive for me because I think whether an image $x\_0$ is good or bad only depends on itself, not its generation process. Could you help me ground this assumption a bit more?
>     ###
>     ### A1
>     It is correct that human judgment typically depends only on the final image $\mathbf{x}\_0$ and not on the specific trajectory that produced it. However, in Eq. 4, the trajectory-level reward $R(\mathbf{x}\_{0:T}, \mathbf{c})$ is introduced as a *modeling choice* to enable more efficient and stable optimization. The key requirement is that the **expected value** of this trajectory reward under the diffusion process aligns with the scalar reward $r(\mathbf{x}\_0, \mathbf{c})$:
>     $$
>     r(\mathbf{x}\_0, \mathbf{c}) = \mathbb{E}\_{\mathbf{x}\_{1:T} \sim p(\cdot \mid \mathbf{x}\_0, \mathbf{c})} \left[ R(\mathbf{x}\_{0:T}, \mathbf{c}) \right].
>     $$
>
>     Importantly, this does **not** assume that human preferences depend on $\mathbf{x}\_{1:T}$; instead, it enables us to decompose the reward across the diffusion steps. This decomposition is beneficial for computational reasons, as it enables gradients to be computed at each denoising step, making the optimization more tractable than backpropagating through the entire diffusion process.
>
>     Moreover, the formulation is flexible. If the learned $R$ chooses to ignore $\mathbf{x}\_{1:T}$ and focus solely on $\mathbf{x}\_0$, it naturally collapses to a standard reward function:
>
>     $$
>     \begin{aligned}
>     r(\mathbf{x}\_0, \mathbf{c}) &= \mathbb{E}\_{\mathbf{x}\_{1:T}} \left[ R(\mathbf{x}\_{0:T}, \mathbf{c}) \right] \\\\
>     &= \mathbb{E}\_{\mathbf{x}\_{1:T}} \left[ R(\mathbf{x}\_0, \mathbf{c}) \right] \\\\
>     &= R(\mathbf{x}\_0, \mathbf{c}).
>     \end{aligned}
>     $$
>
>     This formulation allows flexibility and lets the learning process determine whether information from the full trajectory is beneficial for modeling the reward.
>
> ---
>
> - ### Q2
>     For line 127, the authors say that they parameterize the trajectory reward as similar to Eq. 3. I wonder whether this is a choice made by the authors for math convenience, or perhaps there is some ground truth supporting this parameterization?
>     ###
>     ### A2
>     The parameterization in Eq. 9 is inspired by the formulation used in DPO [A], where the reward is expressed as a log-ratio of policy and reference distributions. As shown in Theorem 1 of DPO, this parameterization can represent any reward function without loss of generality. Therefore, the reward model does not lose any modeling capability with this parameterization. Moreover, this parameterization closely resembles Eq. 3 in our paper, which enables a seamless transition to the closed-form expression for the optimal policy in Eq. 10.
>
>     Thus, the parameterization serves both as a **principled modeling choice** and a **practical tool** that simplifies the derivation while preserving generality.
>
>     [A] "Direct preference optimization: Your language model is secretly a reward model." NeurIPS 2023.
>
> ---
>
> - ### Q3
>     For lines 226-231, I think the text tries to explain why the proposed method, Diffusion-DRO, is outperformed by baselines regarding the CLIP scores. But I don't quite understand the explanations from lines 226-231. Is the reason that Diffusion-DRO only uses the expert demonstrations, a small part of the preference dataset, resulting in a bad text encoder?
>     ###
>     ### A3
>     We would like to clarify that the text encoder used in SD 1.5 shares the same pretrained weights as the one used in CLIP. Since SD 1.5 was trained from scratch on a large dataset, the images it generates naturally align well with the CLIP text encoder, resulting in strong CLIP scores.
>
>     When SD 1.5 is fine-tuned using only a small dataset and a relatively small batch size, the model may slightly deviate from the original distribution. This shift can lead to a mild drop in CLIP score, as the generated images become less aligned with the CLIP encoder.
>
>     This phenomenon can also be observed in models fine-tuned from `SD 1.5 w/ SFT`. For example, in Table 3, `Diffusion-DPO w/ SFT` shows a lower CLIP score compared to the version using the official weights (`Diffusion-DPO`), even though other metrics improve.
>
> ## Clarification of Weakness
>
> ---
>
> - ### Weakness 1
>     "One key concern of the proposed method is that it seems to lose the benefit of DPO methods that are designed to avoid first fitting a reward function and then finding a policy to optimize the reward. Instead, the proposed method in Alg.1 has to first do IRL (Alg.1 line 14) and then do policy optimization (Alg.1 line 16). It is very helpful that Sec.3.4 discusses the frequency of the two alternative steps for IRL and policy optimization. But I wonder whether the alternation between reward and policy learning results in increased computation overhead? Maybe some empirical evaluation of computation could be helpful to address this.
>
>     Also, it is not quite clear to me how the $p\_\phi$ in Alg.1 line 16 is computed. It would be great if the authors could clarify this."
>     ###
>     ### Response to Weakness 1
>     We appreciate the reviewer’s concern regarding the potential overhead of alternating between reward and policy optimization. We would like to clarify that in our framework, the **policy model $p\_\theta$ is always identical to the reward model $p\_\phi$**, as formally derived in Eq. 10. This means that **every update to the reward model simultaneously updates the policy**—no additional optimization step is required for the policy. Line 16 in Algorithm 1 simply copies the parameters ($p\_\theta \leftarrow p\_\phi$) and introduces **no computational overhead**, regardless of the value of $M$. In all of our experiments, we set $M = 1$ so that the policy and reward model remain synchronized at all times. We will add this detail explicitly in Appendix A.2 of the revised version.
>
>     Please note that this design closely mirrors the efficiency advantage of DPO: by training only a single model (the reward model), the generation policy is optimized implicitly. Our proposed Diffusion-DRO thus retains the same **single-model training benefit** as DPO, despite being derived from an IRL perspective. We will revise the paper to make this explanation more explicit and appreciate the reviewer’s feedback on this point.
>
> - ### Weakness 2
>     Another concern is to find demonstration data. Preference data are largely available and easy for humans to provide. But expert demonstrations are hard to obtain. This paper proposes to use the top K images as the expert demonstrations, from a given preference dataset ranked by the preference labels. This makes sense, but introduces a hyperparameter K to tune.
>     ###
>     ### Response to Weakness 2
>     The table below shows the original data used to produce Figure 3 in our paper. As noted in the main text, expert demonstrations are selected based on HPSv2, so we omit HPSv2 scores here.
>
>     As shown, when $K$ ranges from 100 to 1000, Diffusion-DRO consistently outperforms the baselines across most metrics and performs comparably in the remaining cases. While our method introduces a hyperparameter $K$, we find that it performs robustly across a reasonably range of values. It is not necessary to fine-tune $K$ precisely to obtain strong performance.
>     |   Method                 | PickScore | Aesthetic |  CLIP    | ImageReward |
>     |:-------------------------|:---------:|:---------:|:--------:|:-----------:|
>     | Diffusion-DPO            |  21.31    |   6.35    |  0.3334  |   0.7912    |
>     | Diffusion-KTO            |  21.25    |   6.34    |  0.3296  |   0.7636    |
>     | SPIN-Diffusion           |  21.40    |   6.26    |  0.3305  |   0.5619    |
>     | Diffusion-DRO($K=100$)   |  21.55    | **6.46**  |  0.3373  |   0.7876    |
>     | Diffusion-DRO($K=500$)   |**21.54**  |   6.42    |  0.3390  | **0.8452**  |
>     | Diffusion-DRO($K=1000$)  |  21.46    |   6.34    |  0.3409  |   0.8188    |
>     | Diffusion-DRO($K=5000$)  |  21.25    |   6.30    |**0.3439**|   0.7686    |

---

> ### Comment · Reviewer_82HS · 2025-08-04
>
> Thank you for your clarification and extra results! I really appreciate them!
>
> I am still a bit concerned about the requirement of finding high-quality demonstration. On the one hand, it requires some metrics to find good demonstration data. On the other hand, depending on the selection metrics, we might need different amounts of demo data, making it hard to tune the K parameter. Moreoever, from the table provided in the author response, it seems that the improvement of Diffusion-DRO over Diffusion-DPO is quite small. It is unclear how significant that it in reality. I plan to keep the score now.

---

> > ### Author Response · Authors · 2025-08-05
> >
> > Thank you for your thoughtful feedback and for pointing out these important concerns.
> >
> > **On the practicality of collecting high-quality demonstrations:**:
> >
> > We appreciate the concern regarding the challenge of gathering expert demonstrations. In practice, we find that collecting high-quality demonstration data is actually more feasible than collecting large-scale preference pairs. Expert-level images can be sourced from highly-rated examples on public platforms (e.g., images with high user ratings, "Editor's Choice" content, or works from recognized professionals), and do not require exhaustive pairwise labeling. In contrast, preference data collection often involves significant manual effort to construct and curate sufficient pairwise comparisons for training. Our approach only requires a pool of positive examples, which can be curated from existing datasets or popular online resources, making the process less labor-intensive.
> >
> > While tuning the hyperparameter $K$ may affect the absolute scores, our findings show that as long as the demonstrations are representative of high-quality samples, even moderate variations in $K$ can yield competitive results. In practice, a straightforward selection of top-rated images or expert-curated sets is typically sufficient, and does not require fine-grained manual annotation or labeling.
> >
> > **On the magnitude and significance of improvements:**
> >
> > While the numerical gains of Diffusion-DRO over Diffusion-DPO may appear moderate on certain metrics, we would like to emphasize that these improvements are in fact statistically significant. We conducted paired t-tests comparing Diffusion-DRO against the main baselines (SPIN-Diffusion, Diffusion-KTO, and Diffusion-DPO), using the same training runs and evaluation setup as reported in Table 1 of the paper. The results are summarized below. We marked the results of the paired t-tests on the scores of each baseline. Specifically, $^{*}$ indicates *p* < 0.01 and $^{**}$ indicates *p* < 0.001.
> >
> > |   Method       | PickScore  | Aesthetic | CLIP        | ImageReward |
> > |:---------------|:----------:|:---------:|:-----------:|:-----------:|
> > | Diffusion-DPO  |21.31$^{**}$|6.35$^{*}$ |0.3334$^{**}$|0.7912$^{*}$ |
> > | Diffusion-KTO  |21.25$^{**}$|6.34$^{*}$ |0.3296$^{**}$|0.7636$^{**}$|
> > | SPIN-Diffusion |21.40$^{**}$|6.26$^{**}$|0.3305$^{**}$|0.5619$^{**}$|
> > | Diffusion-DRO  |**21.54**   |**6.42**   |**0.3390**   |**0.8452**   |
> >
> > As shown in the table, Diffusion-DRO achieves statistically significant improvements on all reported metrics. In particular, the gains on PickScore and CLIP score are highly significant (*p* < 0.001). This demonstrates that Diffusion-DRO reliably outperforms strong baselines in a reproducible manner.
> >
> > We hope this clarifies that the observed improvements are not only statistically reliable, but also achieved with demonstration data that is practically feasible to collect.

---

> > > ### Comment · Reviewer_82HS · 2025-08-05
> > >
> > > Thank you so much for the extra results! They demonstrate that the proposed method is promising. I will maintain my overall score of 4.

---

### Official Review · Reviewer_NjRJ · 2025-07-02

**Clarity:** 2
**Significance:** 3
**Originality:** 3
**Rating:** 4
**Confidence:** 4

**Summary:**

The paper presents a DPO style preference optimization framework for diffusion models. Different from standard Diffusion-DPO which requires paired preference data, the proposed framework instead the proposed method requires only positive samples for the task. This is achieved by having a margin-based ranking loss over "expert"/highly-preferred samples compared to the samples generated by the policy model for the same prompt. Comparisons on automated scoring metrics on standard prompt benchmarks indicate that the proposed method outperforms related preference optimization methods (e.g. Diffusion-DPO/KTO etc.) on the SD1.5 model.

**Questions:**

While the paper does an excellent job of distinguishing the proposed method from Diffusion-DPO, it might also be a good idea to explicitly have a conceptual comparison with SPIN-Diffusion (at least in the appendix) since it seems closely related as a method.

Lines 205-208 mention that there are 5 generations per prompt and the image with the median Pickscore value is selected. This seems like an interesting choice for evaluation: while seeds have a huge role to play, wouldn't it be a better idea to simply average the results over all generations (and then report std dev)?

Lines 70-73 mentions that while KTO also decouples the positives and negatives, this could introduce semantic biases. However, even after reading the paper, it's not entirely clear why this cannot happen at all for the proposed method either. Could the authors clarify a bit more about this aspect?

The choice of the expert demonstrations is also quite interesting: while Fig. 3 provides an analysis of the number of samples from the Pick-a-Pic dataset, I'm also curious about would having higher-quality images (e.g. the MJHQ-30k images) or captions (e.g. through recaptioning) as a tool to improve the results of the proposed method.

The paper might also want to acknowledge other works on ranking based DPO e.g. LiPO[a]

[a] Liu et al. "LiPO: Listwise Preference Optimization through Learning-to-Rank", NAACL 2025

**Ethical Concerns:**

["NO or VERY MINOR ethics concerns only"]

**Final Justification:**

The authors have provided experiments further strengthening the claims of the paper in the rebuttal (generalizing to the larger SDXL model, different sources of expert demonstrations, and evaluations beyond pure reward models). Based on these, I believe that the paper meets the bar for acceptance.

**Limitations:**

Yes

**Quality:**

2

**Strengths And Weaknesses:**

Strengths:

The framework presented in the paper is quite interesting (i.e comparing expert demonstrations vs policy generations) and to the best of my knowledge has not been explored before, making it a valuable contribution.

Further, the results when compared to prior preference optimization methods (e.g. Diffusion-DPO/KTO, SPIN-Diffusion) appear quite strong making the overall methodology very compelling.

Weaknesses:

One huge drawback of the paper is that it solely relies on the SD1.5 model to make the case for the proposed method. In the last 3 years, there have been several open-source models which have since then provided superior performance (e.g. SDXL, SD3, Flux, SANA etc.). Even preference optimization methods (e.g. Diffusion-DPO etc.) have been applied to some of these methods with public checkpoints (e.g. SDXL with Diffusion-DPO), therefore it might be a good idea to investigate the quality of the proposed method on newer models.

In terms of the evaluation, it mostly relies on win-rates with reward model metrics (e.g. PickScore, ImageReward etc.). As the paper itself mentions (lines 214-215), this could potentially lead to overfitting to a single type of objective. While evaluations on different reward models and the human study is able to address the issue to some extent, different types of evaluations (e.g. object detector based GenEval or VQAScore based GenAI-bench) might also be a way to have more robust comparison.


Overall, I would be inclined to accept the paper, however, I would love to see more robust evaluations along with results on other models before confidently recommending acceptance.

---

> ### Author Rebuttal · Authors · 2025-07-31
>
> We sincerely thank you for your review. Below, we address each of your questions in detail. Several of these questions also overlap with the concerns raised in the weaknesses section, which we address jointly in our responses.
>
> ## Reply to Questions
>
> ---
>
> - ### Q1
>     While the paper does an excellent job of distinguishing the proposed method from Diffusion-DPO, it might also be a good idea to explicitly have a conceptual comparison with SPIN-Diffusion (at least in the appendix) since it seems closely related as a method.
>     ###
>     ### A1
>     We sincerely appreciate the reviewer’s suggestion. While our main comparison focuses on Diffusion-DPO, we agree that SPIN-Diffusion shares conceptual similarities and merits explicit discussion.
>
>     The key distinction lies in the formulation and training paradigm:
>
>     - **Our method** is grounded in a *max-margin inverse reinforcement learning* (IRL) objective (Eq. 7), which directly compares expert demonstrations against policy samples. This formulation leads to a closed-form expression (Eq. 10), where the optimal policy coincides with the reward model. As a result, the min-max optimization reduces to a *single-stage* minimization over the reward model.
>
>     - **SPIN-Diffusion**, by contrast, adopts a *discriminative training* approach. It frames reward modeling as a binary classification task, i.e., distinguishing between expert and generated samples, and relies on alternating updates of the reward and policy models. This requires *multi-stage* training and careful synchronization to ensure stable optimization.
>
>     We believe this conceptual distinction, i.e., *IRL-based ranking vs. discriminative classification*, captures the core difference between the two approaches. We will include a discussion of this comparison in the appendix to help clarify the conceptual differences more explicitly.
>
>     ---
>
> - ### Q2
>     Lines 205-208 mention that there are 5 generations per prompt and the image with the median Pickscore value is selected. This seems like an interesting choice for evaluation: while seeds have a huge role to play, wouldn't it be a better idea to simply average the results over all generations (and then report std dev)?
>     ###
>     ### A2
>     We follow the evaluation protocol adopted by Diffusion-KTO [A], which selects the image with the median Pickscore for pairwise comparisons in automated win rate evaluation (Appendix A.2 of [A]). This also allows us to use the selected images consistently in both automated evaluation and user studies. As a result, the automated win rates can be directly compared to user study results, which improves their reliability.
>
>     It is worth noting that we have provided the mean and standard deviation of preference scores over all generated samples in Table 3 and Table 4 of Appendix C to address the concern about variability across generations. These results show that the propopsed Diffusion-DRO achieves the highest average preference scores across most metrics, with the only exception being a slight gap in CLIP score on the HPSv2 dataset compared to baseline methods.
>
>     [A] "Aligning Diffusion Models by Optimizing Human Utility." NeurIPS 2024.
>
>     ---
>
> - ### Q3
>     Lines 70-73 mentions that while KTO also decouples the positives and negatives, this could introduce semantic biases. However, even after reading the paper, it's not entirely clear why this cannot happen at all for the proposed method either. Could the authors clarify a bit more about this aspect?
>     ###
>     ### A3
>     The concern about semantic bias is valid and worth clarifying further. In **Diffusion-KTO**, positive and negative samples are explicitly decoupled based on a fixed dataset. If the negative set disproportionately contains certain semantic categories (e.g., images of "cats"), the model may inadvertently learn to avoid generating those concepts, even if they are not inherently undesirable. This fixed split makes the model vulnerable to dataset-induced semantic skew.
>
>     In contrast, **our method generates negative samples online** from the evolving policy. This has two advantages: (1) The negative samples naturally reflect the current distribution of the policy, which tends to be more diverse and balanced than a manually curated negative set. (2) The dynamic nature of online sampling reduces the chance of reinforcing static biases, since what constitutes a "negative" evolves over training.
>
>     While no method can entirely eliminate semantic bias, our approach is specifically designed to **reduce its impact** by avoiding hard-coded negative sets. We will clarify this point in the revised version of the paper.
>
>     ---
>
> - ### Q4 + Weakness 1 + Weakness 2
>     The choice of the expert demonstrations is also quite interesting: while Fig. 3 provides an analysis of the number of samples from the Pick-a-Pic dataset, I'm also curious about would having higher-quality images (e.g. the MJHQ-30k images) or captions (e.g. through recaptioning) as a tool to improve the results of the proposed method.
>     ###
>     ### A4
>     We sincerely thank the reviewer for the thoughtful suggestions and for pointing us to valuable resources. Following your comments in both this question and the weaknesses section, we conducted additional experiments using the larger SDXL model fine-tuned on the higher-quality MJHQ-30k [B] dataset. We also adopted the GenEval benchmark [C] as you suggested, to more comprehensively evaluate the model's performance.
>
>     We ranked MJHQ-30k samples using PickScore and selected the top 500 as expert demonstrations. Both `SDXL + SFT` and `Diffusion-DRO` were fine-tuned on this dataset. For reference, we also include the results of the original `SDXL` and `Diffusion-DPO`, the latter of which was fine-tuned on Pick-a-Pic v2. Following the GenEval protocol, each prompt was used to generate four images to ensure stable evaluation.
>
>     | Method            | Colors  | Color Attr. | Single Object | Two Object | Counting | Position |Overall  |
>     |-------------------|:-------:|:-----------:|:-------------:|:----------:|:--------:|:--------:|:-------:|
>     | SDXL              |  87.23  |  23.00      | 99.69         |   78.03    |   43.75  |   11.75  |57.24    |
>     | Diffusion-DPO     |  86.17  |  25.00      | 98.75         |   85.25    |   48.25  | **13.60**|59.50    |
>     | SDXL + SFT        |  88.94  |**27.00**    | 99.75         |   87.68    |   46.50  |   13.20  |60.51    |
>     | Diffusion-DRO     |**90.43**|**27.00**    | **100.00**    | **87.88**  | **50.00**|   12.00  |**61.22**|
>
>     This demonstrates the method’s strong generalization ability to both a larger backbone (SDXL) and a higher-quality dataset (MJHQ-30k). Notably, even when fine-tuning on expert demonstrations in the form of image-text pairs alone, without explicit preference labels, Diffusion-DRO is able to outperform SFT. For example, it achieves a 1.49 percent gain in the "Colors" category and a 3.5 percent improvement in "Counting."
>
>     [B] "Playground v2.5: Three Insights towards Enhancing Aesthetic Quality in Text-to-Image Generation", arXiv.
>
>     [C] "GenEval: An Object-Focused Framework for Evaluating Text-to-Image Alignment." NeurIPS 2023.
>
>     ---
>
> - ### Q5
>     The paper might also want to acknowledge other works on ranking based DPO e.g. LiPO[a]
>
>     [a] Liu et al. "LiPO: Listwise Preference Optimization through Learning-to-Rank", NAACL 2025
>     ###
>     ### A5
>     We sincerely thank the reviewer for pointing out this relevant work. In our current setting, we treat expert demonstrations as binary preferences (i.e., preferred vs. not preferred) without modeling any internal ordering or relative ranking among them. Therefore, our formulation is more aligned with pairwise or margin-based optimization rather than listwise ranking, which is the focus of LiPO [a].
>
>     That said, we agree that incorporating richer ranking information (such as listwise preferences or soft rankings) could offer valuable additional supervision for fine-tuning generative models. We will cite LiPO in the revised version and highlight it as a promising direction for future work that complements our current approach.

---

> > ### Comment · Reviewer_NjRJ · 2025-08-04
> >
> > I thank the authors for taking the time and providing crucial experiments to validate their claims. Based on these results as well as the clarifications, I will raise my score to 4 (borderline accept).

---

> > > ### Author Response · Authors · 2025-08-08
> > >
> > > Thank you very much for your positive feedback and raising the score. We are glad our response addressed your concerns.

---

### Official Review · Reviewer_zy7V · 2025-07-03

**Clarity:** 2
**Significance:** 3
**Originality:** 3
**Rating:** 3
**Confidence:** 4

**Summary:**

This paper introduces Diffusion-DRO, a preference learning framework for text-to-image diffusion models. The method derives a new training objective that is an linear estimation of the denoising losses and it enables fine-tuning diffusion models using only expert demonstrations with online generated examples. Empirically, the method outperforms prior methods by a large margin in terms of win-rates in across various automated metrics.

**Questions:**

1. What is the intuition behind Eq. 7? In line 122, the paper states that "the reward model aims to maximize the margin between the expert and the policy," but this seems inconsistent. If the expert trajectory (left-hand side) is fixed and the reward model is optimized using samples from the policy (right-hand side), wouldn’t this minimize the margin? Could the authors clarify this apparent contradiction?
2. Eq. 9 introduces a new parameterization of the trajectory reward model. What is the intuition behind this formulation? In prior work, similar parameterizations are usually derived from a definition of $R$ and training objective.
3. In line 8 of Algorithm 1, how is the sample $x_t \sim p_\theta$ efficiently obtained? Does this require performing the full denoising process from step $T$ to $t$ ? If so, how is this efficient during training?

I lean toward rejection primarily due to theoretical opacity and limited evaluation scope, though the core idea is novel and the results are promising. I could raise my score with improved clarity and broader experiments.

**Ethical Concerns:**

["NO or VERY MINOR ethics concerns only"]

**Final Justification:**

The author provides more empirical results showing the effectiveness of their method, which could have been a plus. However, the derivation does not come naturally, and the intuition is not well explained; this is a minus. I believe the latter is important to NeurIPS; thus, I lean toward rejection.

**Limitations:**

Yes

**Paper Formatting Concerns:**

No major formatting issues.

**Quality:**

3

**Strengths And Weaknesses:**

## Strengths
1. The method is novel as it replaces the standard non-linear loss with a linear, denoising-based objective for preference optimization in diffusion models, offering a simpler and more interpretable alternative to DPO-style methods.
2. The paper is well written and technically sound, with a clear setup and most part of the derivations are rigorous.
3. The empirical results are strong and show consistent and significant improvements over prior work

## Weaknesses
1. Parts of sections 3.2 and 3.3 are redundant and dense, with limited intuition provided. Though the derivations are complete, it is difficult to follow how the inverse reinforcement learning formulation leads to the final denoising loss (Eq. 7–12). The connection between margin-based objectives and reward-based objective could be better explained.
2. The experiments are limited to SD 1.5, while it is now standard to also evaluate on larger models like SDXL, as done in Diffusion-DPO/SPO. This is important as gains may diminish on stronger base models.

---

> ### Author Rebuttal · Authors · 2025-07-31
>
> We sincerely appreciate your thoughtful review. Below, we address each of your questions in detail. Lastly, we provide clarifications for several concerns mentioned in the weaknesses section that were not explicitly formulated as questions.
>
> ## Reply to Questions
>
> ---
>
> - ### Q1
>     What is the intuition behind Eq. 7? In line 122, the paper states that "the reward model aims to maximize the margin between the expert and the policy," but this seems inconsistent. If the expert trajectory (left-hand side) is fixed and the reward model is optimized using samples from the policy (right-hand side), wouldn’t this minimize the margin? Could the authors clarify this apparent contradiction?
>     ###
>     ### A1
>     Thank you for the insightful question. We would like to clarify the underlying optimization dynamics. Eq. 7 is given as:
>     $$
>     \mathbb{E}\_{\mathbf{c}\sim\mathcal{C},\bar{\mathbf{x}}\_0\sim\mathcal{D}(\mathbf{c})}\Big[r\_{\phi}(\bar{\mathbf{x}}\_0,\mathbf{c})\Big]\ge\mathbb{E}\_{\mathbf{c}\sim\mathcal{C},\mathbf{x}\_0\sim p\_{\mathbf{\theta}}(\mathbf{x}\_0|\mathbf{c})}\Big[r\_{\phi}(\mathbf{x}\_0,\mathbf{c})\Big].
>     $$
>
>     The reward model $r\_\phi$ is trained to assign higher scores to expert demonstrations than to policy samples, thereby maximizing the margin. In contrast, the policy (updated via Eq. 10) implicitly attempts to **close the gap** by generating higher-reward samples. This creates a minimax dynamic similar to IRL.
>
>     In short:
>     - $r\_\phi$ maximizes the margin: $r(\bar{x}\_0) > r(x\_0)$,
>     - $p\_\theta$ reduces the margin by improving sample quality,
>     - The contradiction is resolved by understanding the alternating optimization roles.
>
>     We will revise the text to make this dynamic more explicit.
>
>     ---
>
> - ### Q2
>     Eq. 9 introduces a new parameterization of the trajectory reward model. What is the intuition behind this formulation? In prior work, similar parameterizations are usually derived from a definition of and training objective.
>     ###
>     ### A2
>     We adopt the parameterization in Eq. 9 following the approach used in DPO [A]. As shown in Theorem 1 of DPO, this formulation can represent any reward function (up to an additive constant), ensuring no loss of generality. Importantly, our Eq. 9 mirrors the structure of Eq. 3, enabling a smooth derivation of the closed-form policy in Eq. 10. This consistency simplifies the optimization pipeline by making the reward model and policy model interchangeable under KL-regularized objectives. We believe this formulation balances theoretical soundness and practical convenience, facilitating both analysis and implementation.
>
>
>     [A] "Direct preference optimization: Your language model is secretly a reward model." NeurIPS 2023.
>
>     ---
>
> - ### Q3
>     In line 8 of Algorithm 1, how is the sample $x\_t \sim p\_\theta$ efficiently obtained? Does this require performing the full denoising process from step T to t? If so, how is this efficient during training?
>     ###
>     ### A3
>     In our implementation, we explored two strategies to accelerate the sampling of $x\_t \sim p\_\theta$.
>
>     We first experimented with maintaining a replay buffer that stores intermediate samples $x\_t$ for reuse. In each iteration, only a small portion of the buffer is updated, and full training batches are sampled from the buffer. However, we observed that when the policy is updated every step (i.e., $M = 1$ in line 15 of Algorithm 1), this strategy led to suboptimal performance because the buffer tends to accumulate outdated samples and negatively affects training. As a result, we did not adopt the replay buffer approach in our final implementation.
>
>     Afterward, we considered using more efficient samplers, such as DDIM [B] and DPM-Solver [C]. As mentioned in Section 4 of the paper, we ultimately adopted DPM-Solver++ for faster sampling. Below, we present a comparison between training with the DDPM sampler and DPM-Solver++:
>
>     |Sampler|PickScore|Aesthetic|CLIP|ImageReward|Training Time|
>     |-------|:-------:|:-------:|:--:|:---------:|:-----------:|
>     |DDPM(steps=50)       |21.50|6.48|0.3259|0.7501|46.264 hours|
>     |DPMSolver++(steps=20)|21.54|6.42|0.3390|0.8452|25.708 hours|
>
>     As shown in the table, DPM-Solver++ reduces training time by 44 percent while slightly improving performance across several metrics. We note that this setup was adopted without extensive hyperparameter tuning, suggesting that there is still considerable room for further improving training efficiency.
>
>     [B] "Denoising diffusion implicit models." ICLR 2021.
>
>     [C] "DPM-Solver: A Fast ODE Solver for Diffusion Probabilistic Model Sampling in Around 10 Steps." NeurIPS 2022.
>
> ## Clarification of Weakness
>
> ---
>
> - ### Weakness 1
>     The experiments are limited to SD 1.5, while it is now standard to also evaluate on larger models like SDXL, as done in Diffusion-DPO/SPO. This is important as gains may diminish on stronger base models.
>     ###
>     ### Reply to Weakness 1.
>     Thank you for pointing out this important limitation. In response to a suggestion from reviewer `NjRJ`, we extended our experiments to a larger model and dataset, specifically SDXL and MJHQ-30k [D]. We also adopted the GenEval benchmark [E], as proposed by NjRJ, to more comprehensively assess model performance.
>
>     We ranked MJHQ-30k samples using PickScore and selected the top 500 image-text pairs as expert demonstrations. Both `SDXL + SFT` and `Diffusion-DRO` were fine-tuned on this dataset. For reference, we also report results from the original `SDXL` model and `Diffusion-DPO`, the latter of which was fine-tuned on Pick-a-Pic v2. Following the GenEval protocol, each prompt was used to generate four images to ensure reliable evaluation.
>
>     | Method        |   Colors  | Color Attr. | Single Object | Two Object |  Counting |  Position |  Overall  |
>     | ------------- | :-------: | :---------: | :-----------: | :--------: | :-------: | :-------: | :-------: |
>     | SDXL          |   87.23   |    23.00    |     99.69     |    78.03   |   43.75   |   11.75   |   57.24   |
>     | Diffusion-DPO |   86.17   |    25.00    |     98.75     |    85.25   |   48.25   | **13.60** |   59.50   |
>     | SDXL + SFT    |   88.94   |  **27.00**  |     99.75     |    87.68   |   46.50   |   13.20   |   60.51   |
>     | Diffusion-DRO | **90.43** |  **27.00**  |   **100.00**  |  **87.88** | **50.00** |   12.00   | **61.22** |
>
>     This demonstrates the method’s strong generalization ability to both a larger backbone (SDXL) and a higher-quality dataset (MJHQ-30k). Notably, even when fine-tuning on expert demonstrations in the form of image-text pairs alone, without explicit preference labels, Diffusion-DRO is able to outperform SFT. For example, it achieves a 1.49 percent gain in the "Colors" category and a 3.5 percent improvement in "Counting."
>
>     We hope this additional experiment directly addresses your concern and further demonstrates the scalability and effectiveness of our method in more advanced settings.
>
>     [D] "Playground v2.5: Three Insights towards Enhancing Aesthetic Quality in Text-to-Image Generation", arXiv.
>
>     [E] "GenEval: An Object-Focused Framework for Evaluating Text-to-Image Alignment." NeurIPS 2023.

---

> > ### Author Response · Authors · 2025-08-05
> >
> > Dear Reviewer `zy7V`,
> >
> > Thank you very much for your thoughtful review and for providing a borderline decision. Your feedback is very important to us and could significantly impact the final outcome of our submission.
> >
> > If you have any additional thoughts, clarifications, or questions after reading our rebuttal and the ongoing discussion, we would greatly appreciate your further input. Your perspective is invaluable for us to understand your main concerns more deeply and to improve the quality of our work.
> >
> > Thank you again for your time and for contributing to a fair and thorough review process!

---

> ### Comment · Reviewer_zy7V · 2025-08-05
>
> Thank you for the detailed response and for providing new empirical results. I do appreciate the effort.
>
> However, I still have concerns regarding the theoretical novelty and the underlying intuitions.
>
> - In particular, W1 was not addressed: although the response refers to “W1,” it actually discusses W2 from my review.
>
> - Regarding Q2, the statement that “our Eq. 9 mirrors the structure of Eq. 3” does not, in my view, constitute a strong theoretical contribution. For comparison, while Diffusion-DPO/SPO may share a similar training loss with DPO, the derivations are from the RLHF objective; therefore, simply mirroring an objective’s structure is not sufficient.
>
> I encourage the authors to clarify the derivations and motivations more explicitly. It is still unclear which parts are assumed, which are derived, and which are structural analogies. Addressing this would likely require substantial revision, so at this stage, I lean toward rejection.
>
> Nevertheless, I do appreciate the authors’ additional experiments and the effort put into the rebuttal.

---

> > ### Author Response · Authors · 2025-08-07
> >
> > We apologize for the confusion in our previous response regarding the numbering of weaknesses. Specifically, our response labeled "Reply to Weakness 1" (the SDXL and GenEval experiments) was actually meant to address your Weakness 2 about experiment scope, not Weakness 1. For Weakness 1 (theoretical intuition and derivation), we actually addressed it in our responses to your questions (Q1, Q2, Q3). Below we provide a more explicit and step-by-step clarification, as you suggested:
> >
> > - **Response to W1**:
> >
> >     To clarify your concerns regarding Eq. 7 through Eq. 12, we briefly summarize the core ideas of this part of our method. We begin by restating the objectives for both the policy and reward models:
> >     - Policy optimization
> >         $$\max\_\theta\mathbb{E}\_{{c}\sim\mathcal{C},{x}\_{0:T}\sim p\_{\theta}({x}\_{0:T}|{c})}\Big[R\_\phi({x}\_{0:T},{c})\Big] - \beta\mathbb{D}\_\text{KL}\Big[p\_{\theta}({x}\_{0:T}|{c})\big\Vert p\_{{\theta}\_\text{ref}}({x}\_{0:T}|{c})\Big].\qquad\text{(5)}
> >         $$
> >     - Reward optimization
> >         $$\max\_\phi\mathbb{E}\_{{c}\sim\mathcal{C},\bar{{x}}\_{0:T}\sim\mathcal{D}({c})}\Big[R\_\phi(\bar{{x}}\_{0:T},{c})\Big]  - \mathbb{E}\_{{c}\sim\mathcal{C},{x}\_{0:T}\sim p\_{\theta}({x}\_{0:T}|{c})}\Big[R\_\phi({x}\_{0:T},{c})\Big].\qquad\text{(8)}
> >         $$
> >
> >     - Joint optimization in min-max form
> >     $$
> >     \max\_{\phi}\min\_{\theta}\mathbb{E}\_{{c}\sim\mathcal{C},\bar{{x}}\_{0:T}\sim\mathcal{D}({c})}\Big[R\_{\phi}(\bar{{x}}\_{0:T},{c})\Big] - \mathbb{E}\_{{c}\sim\mathcal{C},{x}\_{0:T}\sim p\_{\theta}({x}\_{0:T}|{c})}\Big[R\_{\phi}({x}\_{0:T},{c})\Big] + \beta\mathbb{D}\_\text{KL}\Big[p\_{\theta}({x}\_{0:T}|{c})\big\Vert p\_{{\theta}\_\text{ref}}({x}\_{0:T}|{c})\Big]
> >     $$
> >
> >     Notably, Eq. 5 closely resembles the RLHF objective used in DPO [29], as shown in Eq. 1:
> >     $$
> >     \max\_{\theta}\mathbb{E}\_{{c}\sim\mathcal{C},{x}\_0\sim p\_{\theta}({x}\_0|{c})}\Big[r({c},{x}\_0)\Big] -\beta\mathbb{D}\_\text{KL}\Big[p\_{\theta}({x}\_0|{c})\big\Vert p\_{{\theta}\_\text{ref}}({x}\_0|{c})\Big],\qquad\text{(1)}
> >     $$It is well established that this objective leads to the optimal policy solution in Eq. 2, which has been widely adopted in prior works [11, 15, 24, 25]:
> >     $$
> >     p\_{\theta}^\*({x}\_0|{c})=\frac{1}{Z({c})}p\_{\theta\_\text{ref}}({x}\_0|{c})\exp\bigg(\frac{1}{\beta}r({x}\_0,{c})\bigg).\qquad\text{(2)}
> >     $$These results further imply the algebraic relationship between reward and policy:
> >     $$
> >     r({x}\_0,{c})=\beta\log\frac{p\_{\theta}^\*({x}\_0|{c})}{p\_{{\theta}\_\text{ref}}({x}\_0|{c})}+\beta\log Z({c}).\qquad\text{(3)}
> >     $$Following the same reasoning of Eq. 2, we can derive the optimal solution for Eq. 5, given any reward function $R\_\phi$:
> >     $$
> >     p\_{\theta}^\*({x}\_{0:T}|{c})=\frac{1}{Z({c})}p\_{\theta\_\text{ref}}({x}\_{0:T}|{c})\exp\bigg(\frac{1}{\beta}R\_\phi({x}\_{0:T},{c})\bigg).\qquad\text{(a)}
> >     $$
> >
> >     Therefore, we parameterize the trajectory-level reward $𝑅$ in a manner analogous to Eq. 3, which leads to Eq. 9:
> >     $$
> >     R\_{{\phi}}({x}\_{0:T},{c}) = \beta\log\frac{p\_{{\phi}}({x}\_{0:T}|{c})}{p\_{{\theta}\_\text{ref}}({x}\_{0:T}|{c})}+\beta\log Z({c}),\qquad\text{(9)}
> >     $$This parameterization is not arbitrary. As discussed in our earlier response to Q2, it retains full generality of reward functions. Just as Eq. 2 and Eq. 3 are algebraically convertible, substituting Eq. 9 into Eq. (a) yields the implied policy distribution:
> >     $$
> >     \hat{p}\_{\theta}({x}\_{0:T}|{c}) = \hat{p}\_{{\phi}}({x}\_{0:T}|{c}),\qquad\text{(10)}
> >     $$ It indicates that policy optimization can be realized by replicating the reward model. This greatly simplifies the training procedure, as it reduces the full optimization to reward-only training using Eq. 8. The next step involves substituting Eq. 9 into Eq. 8 to derive the final loss function. For clarity and brevity, we omit the algebraic derivation here.
> >
> > - **Response to Follow-up Question of Q2**
> >
> >     Building upon the clarification of **W1**, we would like to clarify that our main contribution is not the parameterization in Eq. 9 itself. Rather, we highlight the following two contributions:
> >     1. We introduce a max-margin inverse reinforcement learning framework, and employ a theoretically supported parameterization to simplify policy optimization as model replication, that is, $\hat{p}\_{\theta}({x}\_{0:T}|{c}) = \hat{p}\_{{\phi}}({x}\_{0:T}|{c})$。
> >     2. We observe that the margin maximization objective in max-margin IRL aligns naturally with this parameterization. Due to the subtractive structure of the loss, the $\log Z({c})$ term in Eq. 9 is canceled out in Eq. 11. This makes the optimization process fully tractable.
> >
> > We hope this step-by-step clarification addresses your concerns and makes the derivation and underlying intuition more accessible. Thank you for highlighting the need for clearer exposition—we will further improve this section in the revision.

---

> ### Comment · Reviewer_zy7V · 2025-08-08
>
> Thank you for the detailed response.
>
> I believe Eq. 1,2,3,a are all standard results from prior work.
>
> As for Eq. (9), how do you verify this parameterization? Starting from Eq. 5 and following an argument analogous to Eq. (3), one has $R_\phi=\log \frac{p_\theta}{p_{ref}}$. However, you write $R_\phi=\log \frac{p_\phi}{p_{ref}}$ in Eq. 9 instead and subsequently conclude $p_\theta=p_\phi$ in the end? $p_\phi$ does not seem to arise naturally in this derivation.
>
> Overall, I appreciate the authors' effort and the empirical results demonstrated in the rebuttal. However, I'm not convinced by the derivation and insights. I will keep my current rating.

---

> > ### Author Response · Authors · 2025-08-08
> >
> > The purpose of Eq. 3 is to show the relationship between the reward model $R\_\phi$ and the optimal policy model $p^\*\_\theta$. This leads us to represent the reward model using the form in Eq. 3. Therefore, when training the reward model, it is sufficient to parameterize a probability distribution $p\_\phi$, as shown in Eq. 9. This approach is consistent with the methods used in DPO (Eq. 5 on page 4 of the paper) and Diffusion-DPO (Eq. 7 on page 4 of the paper).
> >
> > After the reward model is trained, we move on to policy optimization. By substituting $R\_\phi$ into Eq. 2, specifically the first equality in Eq. 10, we reach the conclusion $\hat{p}\_\theta = \hat{p}\_\phi$.
> >
> > In summary, we apply Eq. 2 to derive Eq. 3 during reward modeling, and use Eq. 3 to recover Eq. 2 during policy optimization. This supports the conclusion $\hat{p}\_\theta = \hat{p}\_\phi$.

---

> > > ### Comment · Reviewer_zy7V · 2025-08-08
> > >
> > > > it is sufficient to parameterize a probability distribution $p_\phi$, as shown in Eq. 9.
> > >
> > > At this point, $p_\phi=p_\theta$ is an assumed parameterization, not a conclusion.

---

> > > > ### Author Response · Authors · 2025-08-08
> > > >
> > > > We sincerely thank you for raising this concern. We now introduce the optimization step $n$ to distinguish between $\theta\_n$ and $\phi\_n$ at different steps.
> > > >
> > > > * **Training the reward model $p\_{\phi\_n}$ for a given policy at step $n$**
> > > >
> > > >   We adopt the following parameterization:
> > > >
> > > >   $$
> > > >   R\_{\phi\_n}({x}\_{0:T},{c}) = \beta\log\frac{p\_{\phi\_n}({x}\_{0:T}|{c})}{p\_{{\theta}\_\text{ref}}({x}\_{0:T}|{c})}+\beta\log Z({c}), \qquad\text{(9)}
> > > >   $$
> > > >
> > > >   and train $p\_{\phi\_n}$ using the following objective:
> > > >
> > > >   $$
> > > >   \max\_{\phi\_n}\mathbb{E}\_{{c}\sim\mathcal{C},\bar{{x}}\_{0:T}\sim\mathcal{D}({c})}\Big[R\_{\phi\_n}(\bar{{x}}\_{0:T},{c})\Big] - \mathbb{E}\_{{c}\sim\mathcal{C},{x}\_{0:T}\sim p\_{\theta\_n}({x}\_{0:T}|{c})}\Big[R\_{\phi\_n}({x}\_{0:T},{c})\Big]. \qquad\text{(8)}
> > > >   $$
> > > >
> > > >   Note that both the policy model $p\_{\theta\_n}$ and the reward model $\phi\_n$ share the same subscript $n$.
> > > >
> > > > * **Training the policy model $p\_{\theta\_{n+1}}$ for the next step $n+1$**
> > > >
> > > >   The objective is given by Eq. 5:
> > > >
> > > >   $$
> > > >   \max\_{\theta\_{n+1}}\mathbb{E}\_{{c}\sim\mathcal{C},{x}\_{0:T}\sim p\_{\theta\_{n+1}}({x}\_{0:T}|{c})}\Big[R\_{\phi\_n}({x}\_{0:T},{c})\Big] - \beta\mathbb{D}\_\text{KL}\Big[p\_{\theta\_{n+1}}({x}\_{0:T}|{c})\big\Vert p\_{{\theta}\_\text{ref}}({x}\_{0:T}|{c})\Big]. \qquad\text{(5)}
> > > >   $$
> > > >
> > > >   Here, we use the reward model $R\_{\phi\_n}$ trained in the previous step to train the new policy model $p\_{\theta\_{n+1}}$. Eq. 5 admits a closed-form solution, given by Eq. 2. Substituting Eq. 9 into Eq. 2 yields:
> > > >
> > > >   $$
> > > >   \begin{aligned}
> > > >   p^\*\_{\theta\_{n+1}}({x}\_{0:T}|{c}) &=\frac{p\_{\theta\_\text{ref}}({x}\_{0:T}|{c})}{Z({c})}\exp\bigg(\frac{1}{\beta}R\_{\phi\_n}({x}\_{0:T},{c})\bigg) \\\\
> > > >   &=\frac{p\_{\theta\_\text{ref}}({x}\_{0:T}|{c})}{Z({c})}\exp\bigg(\log\frac{p\_{\phi\_n}({x}\_{0:T}|{c})}{p\_{{\theta}\_\text{ref}}({x}\_{0:T}|{c})}+\log Z({c})\bigg) \\\\
> > > >   &=p\_{\phi\_n}({x}\_{0:T}|{c}). \qquad\qquad\qquad\qquad\qquad\qquad\qquad\qquad\qquad\text{(10)}
> > > >   \end{aligned}
> > > >   $$
> > > >
> > > >   Here, the left-hand side corresponds to step $n+1$ while the right-hand side corresponds to step $n$. Therefore, Eq. 10 is not merely an algebraic manipulation of an assumed parameterization but a conclusion derived from the closed-form solution.
> > > >
> > > > We hope this explanation clearly addresses the relationship between $p\_{\theta\_{n+1}}$, $p\_{\theta\_n}$, and $p\_{\phi\_n}$.

---

> ### Comment · Reviewer_zy7V · 2025-08-09
>
> Thank you for the clarification. My point is that, starting from Eq. 9 (which is the parameterization of $p_\phi$), $ p_\phi=p_\theta$. Breaking this into multiple gradient steps does not change anything.  Otherwise, why not use any other parameterization?
>
> Nevertheless, I appreciate the authors' response and the empirical efforts of this work.

---

> > ### Author Response · Authors · 2025-08-09
> >
> > We sincerely appreciate your question.
> >
> > If the training of the policy model (i.e., $p_\theta = p_\phi$) is replaced by gradient updates, the main issue lies in the controllability of the min-max optimization. In particular, gradient updates require intensive hyperparameter tuning to ensure that the policy model always stays at the optimal solution. Without proper tuning, this can lead to poor convergence or even divergence, and it may also increase the training time. Similarly, if the reward model undergoes large updates during training, the policy model may require a long time to catch up with the reward changes, making the tuning of the reward model’s hyperparameters more difficult.
> >
> > In contrast, our proposed method completely removes the need for hyperparameter tuning in the policy model’s training and ensures that the policy model always stays at the optimal solution (Eq. 2 is the closed-form solution). This allows us to focus solely on tuning the reward model’s hyperparameters without having to coordinate with the policy model, which significantly reduces training difficulty and improves stability.
> >
> > We hope this clarification helps you better understand the contribution of our work.

---

### Note · Authors · 2025-08-12

We sincerely thank the reviewers for their valuable time and constructive feedback. Below we summarize the key strengths recognized by the reviewers:

* **Novelty (`zy7V`, `NjRJ`, `82HS`, `Fj6s`)**: Reviewers acknowledged the novelty and sound motivation of our max-margin inverse reinforcement learning formulation. By simplifying the min–max optimization to maximizing the reward gap between the policy model and the expert, our method provides a principled and effective approach to preference alignment.

* **Strong Performance (`zy7V`, `NjRJ`, `82HS`, `Y9sn`, `Fj6s`)**: Our approach outperforms baselines on most human preference score win rates. This is supported by fair and sufficiently sized user studies, which complement and reinforce the strength of our results.

* **Fair and Transparent Experiment Setup (`zy7V`, `Fj6s`)**: We clearly report all experimental and user study details, along with releasing our code, ensuring that our strong results are established on a fair basis.

We also appreciate the high-quality feedback and suggestions, and we summarize below the clarifications we provided during the rebuttal:

* **Effectiveness on Larger Models (`zy7V`, `NjRJ`, `Fj6s`)**: In response to reviewer requests, we trained on SDXL with the MJHQ-30k dataset and evaluated using GenEval. Diffusion-DRO outperforms SFT, achieving an overall relative improvement of 7.0%.

* **Hyperparameter Robustness (`82HS`, `Fj6s`)**: We conducted ablation studies on hyperparameter robustness, showing effective ranges for both $m$ and top-$K$ where our method consistently outperforms baselines. Additionally, we provided paired t-tests demonstrating that the reported improvements in our paper are statistically significant $(p<0.01)$.

* **Reward Parameterization (`zy7V`, `82HS`)**: We provided detailed explanations showing that our reward model parameterization is without loss of generality and facilitates deriving a closed-form solution for the policy model. We also clarified how reformulating the min-max optimization as a minimization benefits hyperparameter tuning and training stability.

We are encouraged by the opportunity to address the reviewers’ questions and clarify their concerns. We sincerely thank the reviewers again for their thoughtful and valuable feedback.

---

### Decision · Program_Chairs · 2025-09-17

**Decision:**

Accept (poster)

**Comment:**

This paper proposes a method for finetuning diffusion models called Diffusion Denoising Ranking Optimization (Diffusion-DRO). This method does not require paired preference data like DPO, and instead relies on a set of expert demonstrations (high quality images) that serve as positive examples for finetuning. The authors derive Diffusion-DRO from the perspective of max-margin inverse RL. They propose a simple update formula and provide interpretations in terms of its relationship to SFT and DPO.

Empirically, the authors use Diffusion-DRO to finetune Stable Diffusion 1.5, using the 500 top-scoring training examples from Pick-a-Pic v2 as ranked by PickScore and HPSv2 metrics, as the expert demonstrations. They report performance on Pick-a-Pic v2 and HPDv2 using PickScore, HPSv2, Aesthetic score, CLIP score, and ImageReward metrics. They show that Diffusion-DRO outperforms the baselines on most metrics in terms of automated win rates.

The paper also compares to SFT-augmented versions of baselines, by first optimizing the base model with SFT on the same set of expert demonstrations, and then applying a given finetuning algorithm such as DPO. This seems like a nice way to ensure fair comparisons and show the benefit of Diffusion-DRO.

The authors also perform a user study that shows that Diffusion-DRO is preferred to two good baselines (Diffusion-KTO w/ SFT and Diffusion-DPO w/ SFT). They also provide an ablation over the number of expert demonstrations, where some metrics increase and some decrease given more demonstrations.

Reviewer zy7V found the idea to be novel and the results promising, and raised concerns regarding the intuition of the derivation and the scope of the experiments. The reviewer still had concerns regarding the derivation after the rebuttal, but the new results the authors provided on SDXL were helpful to show the effectiveness of the method.

Reviewer NjRJ found the framework novel and interesting, with good empirical results compared to baselines. This reviewer was concerned that the experiments used only Stable Diffusion 1.5 and focused on win-rates with reward model metrics. The rebuttal addressed these concerns by providing results using SDXL, a different source of expert demonstrations, and different evaluation metrics using the GenEval benchmark. Regarding the evaluation using automated win rates, the reward model scores, with means and standard deviations, are provided in Appendix Tables 3 and 4, and they also support the claim that Diffusion-DRO outperforms the baselines.

Reviewer 82HS found the paper to be well-motivated and well-written, and the empirical results to be good. This reviewer raised questions regarding some derivations, comparisons to DPO, and the selection of expert demonstrations. These results were partially addressed by the rebuttal with new results.

Reviewer Y9sn found the paper to have good performance, and had questions regarding the theoretical framework. The rebuttal resolved all of this reviewer’s concerns.

Reviewer Fj6s found the paper to be novel, clearly written, and containing comprehensive experiments. This reviewer raised concerns regarding the generalizability of Diffusion-DRO to larger models, and the rebuttal provided results on SDXL.

Overall, Diffusion-DRO is a well-motivated method that performs well compared to baselines. The perspective from inverse RL is interesting, the paper is clearly written. I think it would be a useful contribution for the community.